# Universal paramyxovirus vaccine design by stabilizing regions involved in structural transformation of the fusion protein

Johannes P. M. Langedijk [1,3], Freek Cox[1], Nicole V. Johnson [2], Daan van Overveld [1], Lam Le[1], Ward van den Hoogen [1], Richard Voorzaat [1], Roland Zahn [1], Leslie van der Fits[1], Jarek Juraszek [1], Jason S. McLellan [2] & Mark J. G. Bakkers [1,3] ✉

The *Paramyxoviridae* family encompasses medically significant RNA viruses, including human respiroviruses 1 and 3 (RV1, RV3), and zoonotic pathogens like Nipah virus (NiV). RV3, previously known as parainfluenza type 3, for which no vaccines or antivirals have been approved, causes respiratory tract infections in vulnerable populations. The RV3 fusion (F) protein is inherently metastable and will likely require prefusion (preF) stabilization for vaccine effectiveness. Here we used structure-based design to stabilize regions involved in structural transformation to generate a preF protein vaccine antigen with high expression and stability, and which, by stabilizing the coiled-coil stem region, does not require a heterologous trimerization domain. The preF candidate induces strong neutralizing antibody responses in both female naïve and pre-exposed mice and provides protection in a cotton rat challenge model (female). Despite the evolutionary distance of paramyxovirus F proteins, their structural transformation and local regions of instability are conserved, which allows successful transfer of stabilizing substitutions to the distant preF proteins of RV1 and NiV. This work presents a successful vaccine antigen design for RV3 and provides a toolbox for future paramyxovirus vaccine design and pandemic preparedness.

The *Paramyxoviridae* family comprises enveloped, negative-sense RNA viruses possessing a non-segmented genome. This group encompasses numerous human endemic viruses that hold significant medical importance, such as measles virus, mumps virus, human rubulavirus types 2 and 4, and human respirovirus types 1 and 3 (human RV1 and RV3, previously known as human parainfluenzavirus type 1 and 3)[1–4]. Additionally, it includes zoonotic viruses, namely Nipah virus (NiV) and Hendra virus, which exhibit high case-fatality rates upon spillover into humans[5–10]. RV3 generally induces mild disease but can also cause severe respiratory tract infections, particularly in vulnerable populations such as infants, the elderly, and immunocompromised individuals[11–14]. RV3 remains a leading cause of hospitalizations associated with respiratory illness. Presently, there are no approved vaccines or therapeutics available specifically targeting RV3[15].

The first steps of the viral lifecycle are attachment to the host cell and penetration of the cellular membranes to gain access to the cytosol where replication occurs. RV3 has two surface glycoproteins that cooperate to mediate these steps: the hemagglutinin-neuraminidase protein (HN) and fusion protein (F). HN performs three functions during the infection process: (i) sialic acid-binding and (ii) receptor-destruction, both performed by the HN head domain, and (iii) activation of the F protein for membrane fusion by the HN stalk

[1]Janssen Vaccines & Prevention BV, Leiden, The Netherlands. [2]Department of Molecular Biosciences, The University of Texas at Austin, Austin, TX, USA. [3]Present address: ForgeBio, Amsterdam, The Netherlands. ✉e-mail: mbakkers@forge-bio.com

region[16–21]. F is a trimeric, class I fusion protein present on the viral surface in a metastable conformation. Synthesized as a single polypeptide chain, it undergoes proteolytic processing by host cell proteases like Transmembrane Protease Serine 2 (TMPRSS2) to generate the F2 and F1 subunits and free the fusion peptide to attain its fusion-competent state[22]. F mediates the fusion of viral and host cell membranes through a large-scale conformational change from the metastable prefusion (preF) to the highly stable postfusion (postF) conformation[23]. Previously, the preF and postF structures of multiple paramyxoviruses have been elucidated, including those of SV5/PIV5, measles virus, Newcastle disease virus, NiV, Hendra virus, Langya virus and RV3[24–32]. The protein constructs used for these studies most often consisted of a soluble F ectodomain fused to the GCN4 trimerization domain (Fig. 1A)[24]. These structural studies have shed light on the conformational changes occurring during the fusion process. Fusion is a multi-step process thought to start with the dissociation of the heptad repeat B (HRB; residues 447-484) coiled-coil structure, followed by movement of domains and repositioning of domain DIII, leading to assembly of the heptad repeat A (HRA; residues 129-192) coiled-coil and release of the fusion peptide that is sandwiched between domains DII and DIII. Finally, the HRB helices fold back and

dock on the HRA coiled-coil forming a 6-helix bundle, thereby finalizing membrane merger[23,33,34].

Stabilizing class I fusion proteins in their prefusion conformation has proven effective for developing safe, effective and recently licensed vaccines against SARS-CoV-2 and respiratory syncytial virus (RSV)[35–38]. These vaccines ushered in a new era of structure-based vaccine design and highlight the importance of engineering strategies to create stabilized prefusion proteins. One of the most widely applicable strategies, the use of C-terminal heterologous trimerization domains (e.g., T4 fibritin foldon, GCN4), has been successfully used to trimerize the labile ectodomains of multiple fusion proteins. The introduction of prolines in regions with alpha-helical propensity, most notable in the HRA hinge, has also worked to stabilize a variety of fusion proteins[36,39–41]. These approaches, along with other stabilizing modifications, e.g., introduction of disulfide bonds, cavity-filling substitutions and removal of buried polar residues, have been applied to RSV F, Ebola GP, Lassa GP, HIV-1 Env, RV-3 F, HMPV F, influenza HA and various coronavirus Spike proteins[30,39–46]. In case of RV-3 F, preF stabilization was previously achieved by fusing the C-terminus to a GCN4 trimerization domain, and introduction of two disulfides, Q162C-L168C and I213C-G230C, in combination with two cavity-filling

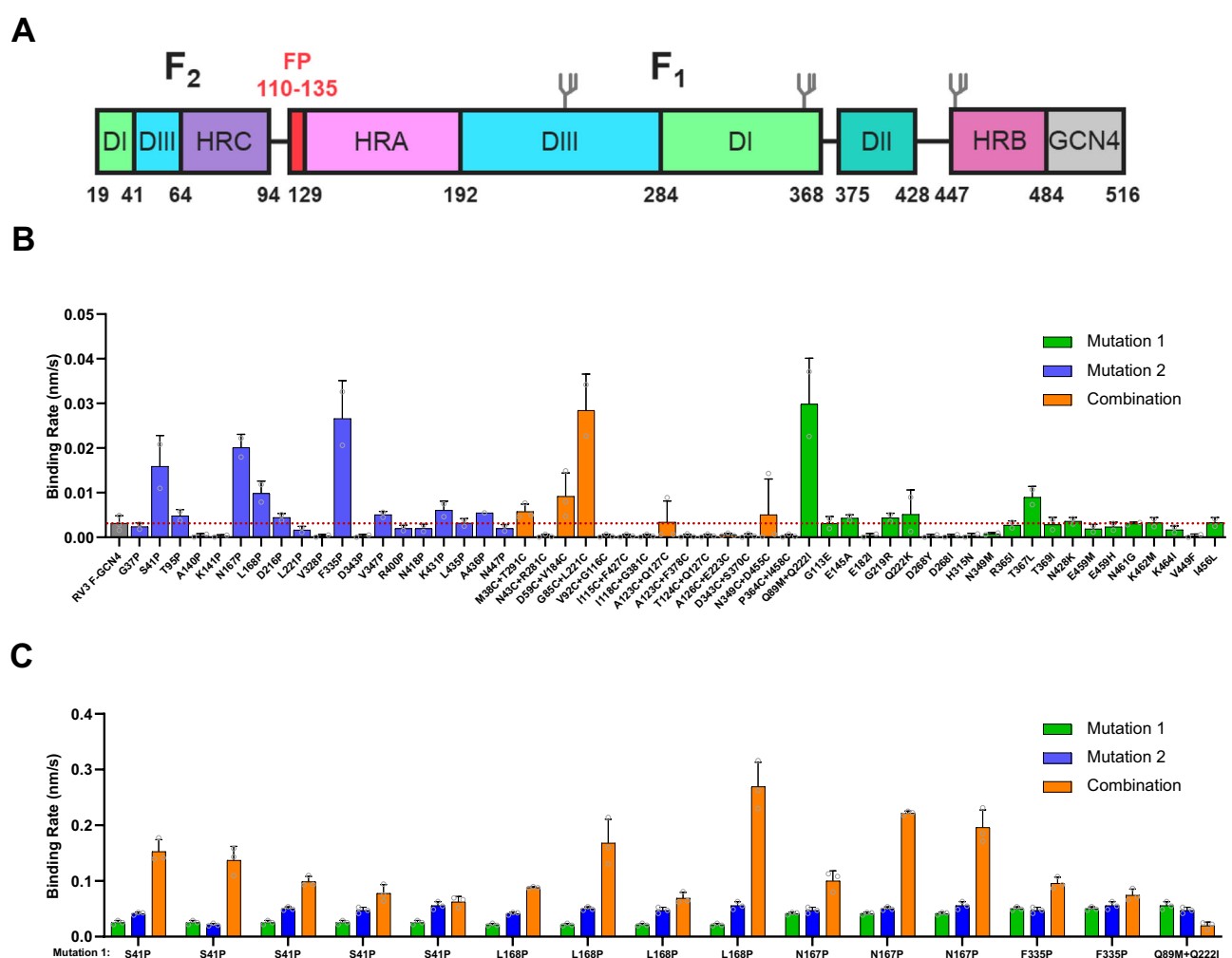

**Fig. 1 | Identification of RV3 preF-stabilizing substitutions. A** Schematic of the RV3 F ectodomain used to screen for stabilizing substitutions in the head domain. The domain organization is color-coded, with start and end residue numbers indicated. Location of N-linked glycosylation sites are shown. **B, C** Expression of preF protein variants in supernatant as measured by BioLayer Interferometry (BLI) using immobilized PIA174. The initial slope, V0, at the start of binding is plotted as

the average of three independent transfections; shown is the mean with error bars representing the standard deviation (SD). **B** Individual substitutions were tested and are colored by design feature. The dotted red line indicates the binding rate of the backbone construct RV3 F-GCN4. **C** Combinations (orange) of promising stabilizing substitutions of (**B**) were tested and compared to the individual substitutions (green and blue).

substitutions, A463V and I474Y[30]. Broadly applicable immunogen design approaches are crucial for pandemic preparedness as they enable rapid design of prefusion antigens for emerging viruses, not only for vaccines but also for structural studies and assays. Here we employed a structure-based design strategy to generate an RV3 F-based vaccine candidate in the desired preF conformation that has high expression and excellent thermal and long-term stability and that does not require a heterologous trimerization domain and will therefore not induce off-target responses. Subsequent in vivo immunogenicity assessment in both naïve and pre-exposed mice demonstrated the candidate's ability to induce robust neutralizing antibody responses as well as to boost recall responses. Moreover, in contrast to the postF protein, the preF candidate provided protection in a naïve cotton rat challenge model. Furthermore, a subset of the substitutions identified in this study can be used to stabilize the prefusion conformation of various other paramyxovirus F proteins. Overall, our approach showcases the successful development of a vaccine candidate with desirable attributes and contributes valuable insights into the stabilization of the preF conformation across diverse paramyxoviruses, underscoring its significance for future vaccine design and pandemic mitigation endeavors.

## Results

### Identification and combination of stabilizing substitutions in the head domain

To screen for substitutions that increase stability and/or expression of the prefusion conformation of RV3 F, single or double amino acid substitutions were introduced in plasmids that code for the RV3 F ectodomain fused to the helical GCN4 trimerization domain in register with the heptad repeat of the C-terminal stem region followed by a C-tag (RV3F-GCN4; Fig. 1A)[24,30]. Individual substitutions fall into three categories: (i) prolines to prevent helix formation or stabilize loops, (ii) disulfide bonds to lock refolding regions, and (iii) removal of buried polar residues to improve packing. Variants were designed using Rosetta based on PDB ID 6MJZ[30] and screened in 96-well format in Expi293F cells and analyzed by Biolayer Interferometry (BLI) using Octet with prefusion-specific antibody PIA174 (Fig. 1B), a monoclonal antibody that binds at the preF apex, along the 3-fold symmetry axis[30]. Three days after transfection, the cell culture supernatants of RV3-GCN4 and 54 F variants were tested for PIA174 binding using BLI. The BLI signal can be influenced by differences in PreF protein stability, differences in expression level, or substitutions in the PIA174 epitope. As to exclude the latter option, no substitutions were screened in or near the epitope of PIA174. Differentiating between effects on preF stability and preF expression is more difficult in this screening approach. However, for vaccine manufacturing, both qualities are important and therefore, any substitutions that increase the BLI signal are of potential interest. In subsequent evaluation rounds using purified protein, the impact on stability of the combination of mutations is more accurately tested.

RV3 F-GCN4 showed very low PIA174 binding, reflecting low stability of wildtype F. Nine out of 19 proline substitutions increased PIA174 binding, with 4 (S41P, N167P, L168P and F335P) showing an increase of 5.1-, 6.4-, 3.1- and 8.5-fold, respectively. S41P is in a long ß-strand that connects domains DI and DIII. N167P and L168P are in the surface-exposed loop of the β4-β5 hairpin of HRA, and F335P is in a surface-exposed loop in domain DI. Two out of 21 substitutions to improve packing or remove buried hydrophilic residues were successful, with T367L giving a 3-fold increase and the Q89M/Q222I double substitution providing a 10-fold increase in PIA174-binding. Five out of 14 disulfides were allowed by the protein, with G85C-L221C improving preF trimer expression 10-fold.

Potential stabilization synergy was evaluated by pairwise combination of stabilizing substitutions. Six single or double stabilizing substitutions were selected for this: four proline substitutions (S41P,

N167P, L168P and F335P), the combination substitution to remove buried hydrophilic residues (Q89M + Q222I) and the most successful disulfide (G85C + L221C). RV3 preF in supernatant was measured using BLI with immobilized PIA174 (Fig. 1C). Combinations involving either N167P or L168P with S41P, Q89M/Q222I, or F335P were beneficial. Of note, the disulfide G85C-L221C was antagonistic with the space-filling pair Q89M/Q222I, which likely arises from L221C and Q222I being neighboring residues.

### Stabilization of the F trimer without support of a heterologous trimerization domain

Next, the inherently labile RV3 HRB domain (residues 447–484) that forms the stem (residues 453–484), was stabilized to obtain a preF trimer without a heterologous trimerization domain, which may be immunogenic and elicit non-protective antibodies. Since the separation of the HRB domain in the stem is an early step in the refolding process, the HRB interface in the prefusion conformation must be suboptimal. Indeed, whereas typical heptad repeats of helical coiled-coils prefer hydrophobic residues like Val, Leu or Ile at the first and fourth positions (a and d), the RV3 F stem contains suboptimal small polar Ser residues at two d positions of the heptad (positions 470 and 477) (Fig. 2A, B). An RV3 F variant with three stabilizing substitutions in the head domain (Q89M + Q222I and L168P) gave the highest trimer expression in presence of GCN4 (Fig. 1C), but only showed low expression of F monomers when the GCN4 trimerization domain was removed, as determined by analytical size-exclusion chromatography (SEC) on cell culture supernatant of transfected Expi293F cells (Fig. 2C, blue line). When the 470 and 477 positions were optimized by substitution of the individual or combined Ser residues for Val residues to allow more favorable interactions (Fig. 2A, B), it resulted in a sharp increase in trimer expression in each case (Fig. 2C).

Next, the effect of S470V and S477V was measured in an F variant that contained two additional head-stabilizing substitutions (S41P and N167P) on top of Q89M + Q222I and L168P. Although this further-stabilized variant already formed trimers without stem stabilization, it displayed a shorter retention time on analytical SEC, reflecting a higher hydrodynamic radius of the trimer and thus a more open quaternary structure (Fig. 2D). When S470V and S477V were introduced into this construct, the retention time increased, implying a more compact structure. Also, the thermal stability of this more compact 'closed' trimer was increased as it remained stable after exposure to 50 °C and 60 °C for 15 min (Fig. 2E). The trimer variant that only contained the five head-stabilizing substitutions aggregated after temperature increase. Although S477V, and especially S470V, increased thermal stability at 50 °C, only the combination of both substitutions led to a stable preF trimer that was fully retained after exposure to 60 °C.

### Impact of stabilizing substitutions on fusogenicity

Elucidation of the effects of the stabilizing substitutions on the fusogenicity of the F protein improves understanding of the fusion mechanism and holds significance in evaluating the feasibility of extending the stabilization strategy to RNA-based or alternative vector-mediated vaccine modalities, commonly dependent on the synthesis of integral, membrane-bound constructs. Therefore, we developed a cell-cell fusion assay in adherent HEK293 cells in which we can visualize the formation of multi-nucleated cells, also known as syncytia. Plasmids encoding the full-length F protein sequence of the JS strain of RV3 were co-transfected with HN- and mScarlet-encoding plasmids to facilitate F protein activation and visualization of syncytia, respectively.

To assess the effect of the stabilizing substitutions, membrane-bound wildtype and variants were transfected and imaged 18 h post-transfection (Fig. 3). Whereas the wildtype F protein induced prominent syncytia, most of the tested variants failed to trigger syncytia

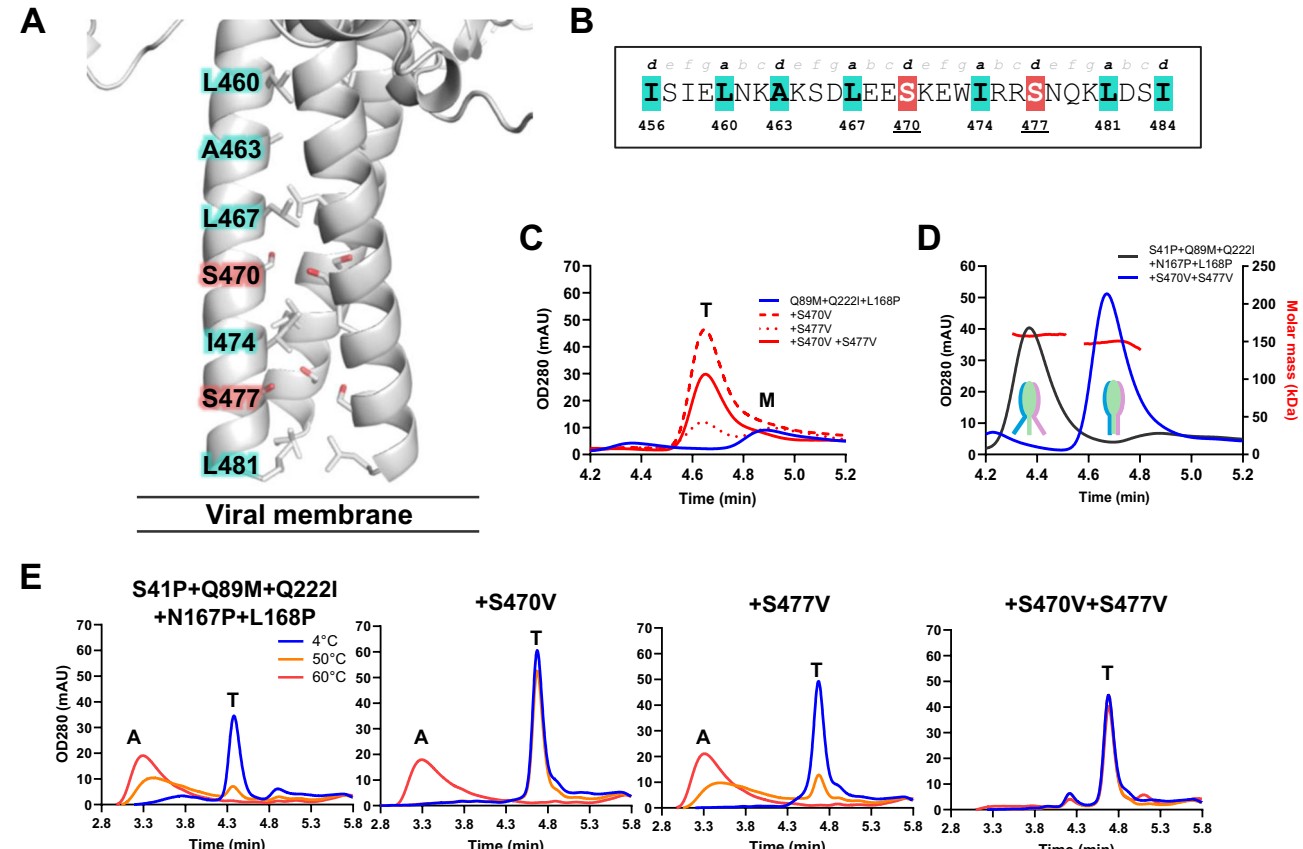

**Fig. 2 | Stabilization of the RV3 HRB region. A** Cartoon representation of HRB from PDB ID 6MJZ. The heptad register is shown in sticks, with the serines at positions 470 and 477 highlighted in red. **B** The RV3 F HRB heptad register indicating the suboptimal residues at positions 470 and 477. **C** Analytical SEC trace of F variants with a minimally stabilized head domain (Q89M, Q222I and L168P) and with or without HRB stabilization in supernatant. The trimer (T) and monomer (M) peaks are indicated. **D** Analytical SEC-MALS trace of F variants with a stabilized head domain with or without HRB stabilization in supernatant. The molar mass as determined by MALS at peak max of the trimer are indicated. A cartoon to visualize the presumed opening of HRB is shown. **E** Stability of indicated F variants using analytical SEC of supernatants after 15 min incubation at 4 °C (blue line), 50 °C (orange line), or 60 °C (red line). The trimer (T) and aggregate (A) peaks are indicated.

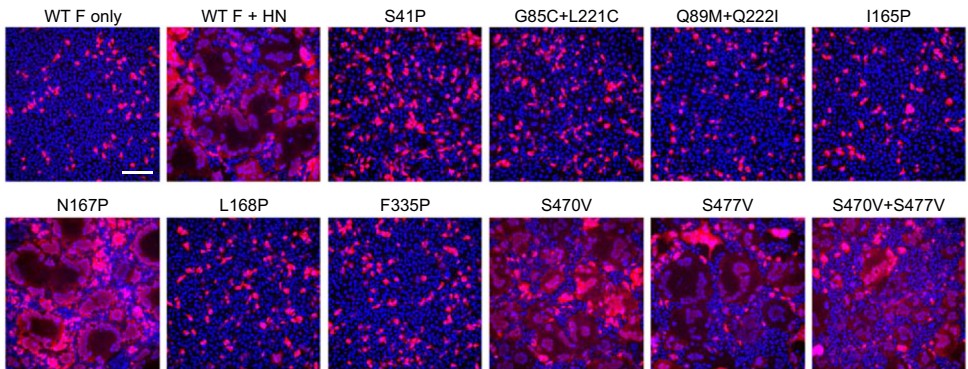

**Fig. 3 | Fusogenicity of full-length RV3 F variants.** Cell-cell fusion assay using HEK293T cells transiently transfected with the F protein of the RV3 JS strain, or variants thereof, in a 15:1 ratio to HN carrying a H552Q substitution, to allow fusion activation, and mScarlet to mark the cytosol of transfected cells in red. Cell nuclei were visualized with Hoechst in blue. Syncytia can be recognized by the dilution of the mScarlet signal after fusion with non-transfected cells, and the clustering of nuclei. Scale bar represents 100 μm. Experiment was performed twice with similar outcome, data shown are from the same, representative experiment.

formation (Fig. 3). Notably, N167P retained the ability to induce syncytia formation, whereas the adjacent substitution L168P did not. The stem-stabilizing substitutions S470V and S477V individually did not impede fusion but their combination did seem to reduce syncytia formation slightly. To confirm that the lack of fusion observed for the stabilizing substitutions was not caused by a lack of F

expression, wildtype and variant proteins were concomitantly expressed in Expi293F cells and subsequently analyzed using fluorescence-activated cell sorting (FACS) with antibody PIA174 (Supplementary Fig. 1). FACS analysis confirmed that the levels of preF expression were comparable between the wildtype construct and the variants.

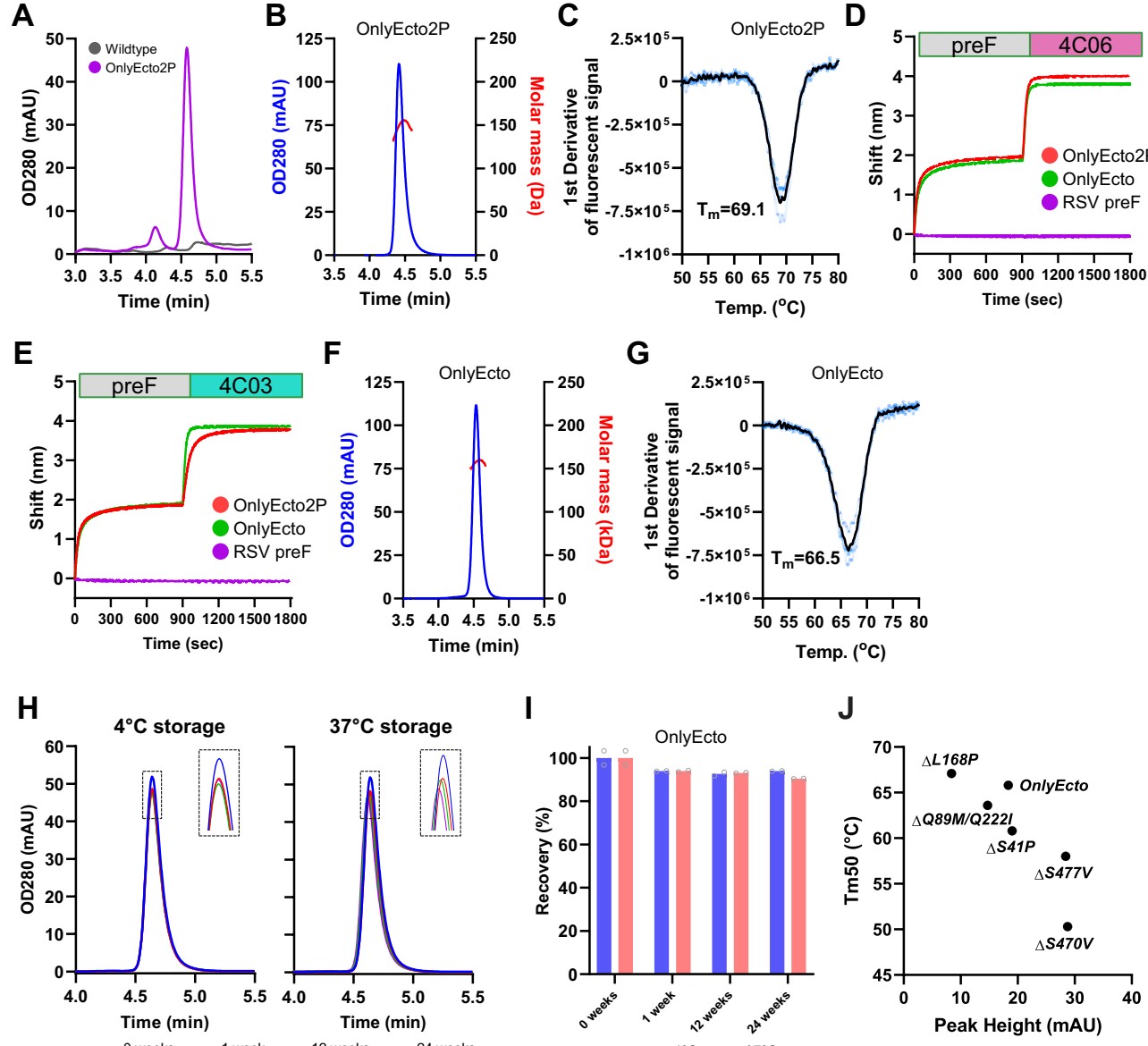

**Fig. 4 | Purification and characterization of RV3 preF lead candidates.**
**A** Analytical SEC trace of wildtype (gray) and stabilized OnlyEcto2P (purple) in supernatant. **B** SEC-MALS trace of purified OnlyEcto2P. **C** Melting temperature as determined by DSF of purified OnlyEcto2P of (**B**). The first derivative of the fluorescence signal is plotted. BLI using kinetic Octet with PIA174 immobilized to anti-human Fc sensors (not shown), followed subsequently by capture of purified preF variants (0–900 s phase) and association of sdAb's 4C06 (**D**) and 4C03 (**E**)

(900–1800 s phase). **F** SEC-MALS trace of purified OnlyEcto. **G** Melting temperature as determined of OnlyEcto as in (**C**). **H** Analytical SEC trace of purified OnlyEcto after storage at either 4 °C (*left panel*) or 37 °C (*right panel*) for up to 24 weeks. **I** Percentage trimer recovery of (**H**). Shown is the average of two replicates. **J** Expression and melting temperature of OnlyEcto and back-substitutions as measured by analytical SEC and DSF on crude cell supernatant.

## Stabilized RV3 F is in the prefusion conformation and retains neutralizing epitopes

Next, a variant with 8 of the most promising stabilizing substitutions (S41P, Q89M, Q222I, N167P, L168P, F335P, S470V and S477V), as based on preF expression (Fig. 1B), compatibility with each other (Fig. 1C) and ability to prevent cell-cell fusion (Fig. 3), termed 'OnlyEcto2P', was expressed in a side-by-side comparison to a non-stabilized wildtype protein (Fig. 4A). Trimer expression was not detected for the unmodified wildtype ectodomain, underlining the importance of stabilizing substitutions. The stabilized variant expressed as a trimer, with a retention time of 4.58 min. This design was purified and analyzed using Size Exclusion Chromatography coupled to Multi-Angle Light Scattering (SEC-MALS) (Fig. 4B). The measured masses for this preF protein (156 kDa) correspond well to

the expected mass of 156.3 kDa, thus confirming that the protein is trimeric. The melting temperature of the purified protein was determined using differential scanning fluorimetry (DSF), showing a $Tm_{50}$ for OnlyEcto2P of 69.1 °C (Fig. 4C). Antigenicity of the purified proteins was evaluated using mAb PIA174 and single-domain antibodies (sdAb) 4C03 and 4C06 which bind to non-overlapping epitopes around the equator of the head domain. Purified OnlyEcto2P was captured with PIA174 immobilized to anti-human Fc sensors and exposed to either 4C06 (Fig. 4D) or 4C03 (Fig. 4E) sdAb. The binding rate of 4C03 was slower than expected when compared to 4C06. Since the F335P stabilizing substitution is surface-exposed and in the center of the 4C03 epitope (Supplementary Fig. 2) a variant (OnlyEcto) without the C-tag and in which residues F335P and another surface-exposed residue, N167P, were reverted to wildtype

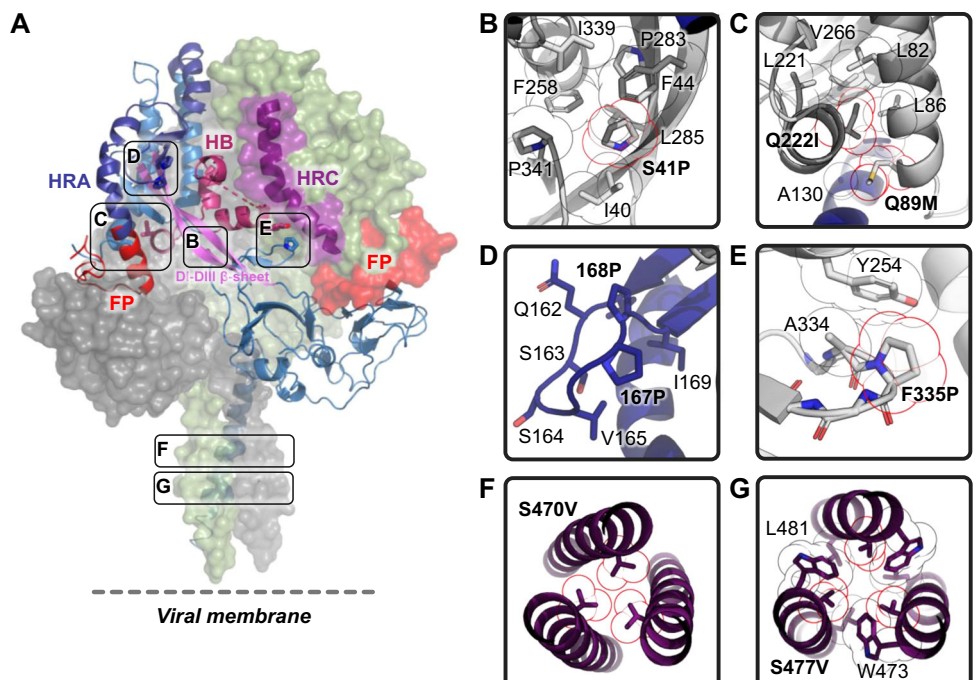

**Fig. 5 | Structure of stabilized RV3 preF. A** Structure of OnlyEcto2P. Two monomers are shown as surface representation, and one as cartoon. Different structural elements are color-coded and indicated. Boxes refer to stabilizing substitutions and refer to Fig. 5B–G. (HRA heptad repeat A (dark blue), HRC: heptad repeat C (purple), HB: helical-bundle (maroon), FP: fusion peptide (red), DI-DIII β-sheet (pink)) (**B**) P41 buried in the hydrophobic pocket forming hydrophobic contacts involving sequence distant residues. **C** Space filling double substitution Q89M + Q222I in its aliphatic pocket formed by three helices. **D** The N167P and L168P substitutions in the DI β-hairpin. **E** Turn stabilizing F335P substitution. **F** S470V on the threefold axis of the stalk. **G** S477V forming an unusual coiled-coil structure with neighboring W473 and L481 and a central water channel.

was purified. Analysis using SEC-MALS showed a symmetric peak at the expected retention time (Fig. 4F) and DSF indicated a $Tm_{50}$ of 66.5 °C (Fig. 4G). Importantly, OnlyEcto showed similar 4C06 binding compared to OnlyEcto2P, but an increase in binding rate to 4C03 (Fig. 4D, E). In addition, purified OnlyEcto protein remained stable during long-term storage at both 4 °C and 37 °C for up to 24 weeks (Fig. 4H, I).

To evaluate the contribution of the introduced stabilizing substitutions in OnlyEcto, variants with back-substitutions were tested for expression and stability (Fig. 4J, Supplementary Fig. 3). S41P, S470V and S477V are important for maintaining preF stability, with S470V providing a very strong improvement in thermal stability of +15.5 °C, suggesting that the melting temperature corresponds to opening of the F protomers. The combination of Q89M/Q222I is contributing to both preF expression and thermal stability. In contrast, L168P leads to a reduction in thermal stability of −1.3 °C, but the trimer expression increases twofold.

### Structure of stabilized RV3 preF

To further verify the prefusion conformation and to understand how the substitutions stabilize prefusion F, a cryo-EM structure of OnlyEcto2P in complex with sdAb 4C06 was determined (Fig. 5A). This design was chosen because it has additional stabilizing substitutions compared to the lead candidate OnlyEcto, and a structure of those substitutions would be informative. A dataset of 2061 images was collected and processed, resulting in a 3.1 Å resolution 3D reconstruction (Supplementary Figs. 4–6, Supplementary Table 1). The refined model has a root-mean-square deviation of 2.7 Å for all backbone atoms compared to the previously solved structure[47]. Several regions are resolved that were previously poorly resolved or disordered, particularly the HRB stem, which shows an extension of the triple helix with additional density for residues 473–484, as well as the

217–224 helix in the helical bundle (HB) which was previously disordered or random coil[30,47].

The structure provides insight into the mechanism behind the stabilizing substitutions introduced in the RV3 preF design. S41P, situated at a bend in the extended β-sheet structure between DI and DIII, occupies an intra-protomeric pocket formed by four sequence-distant groups of residues: 40 + 44, 258, 283–285, and 340–341 (Fig. 5B). This hydrophobic pocket accommodates a poorly fitting S41 in the wild-type structure, which is better packed by the substitution to proline. The Q89M + Q222I substitutions (Fig. 5C) stabilize a region of the protein that was partially disordered in previous structures (e.g., PDB ID 8DG8[47]) and create a highly organized hydrophobic cluster, which encompasses three distinct helices. The N167P + L168P substitutions are within the β4-β5 hairpin loop (Fig. 5D) which constitutes a segment of the HRA region that transitions into an extended helical bundle in the post-fusion state and serves to rigidify the loop due to the restricted torsion angles of proline residues. The F335P substitution (Fig. 5E) is in a surface-exposed β-hairpin and eliminates the unusually solvent-exposed hydrophobic F335 while simultaneously reinforcing the turn structure, thus exerting a substantial stabilizing effect. The HRB substitution S470V (Fig. 5F) fills the small cavity in between the HRB helices with space-filling, hydrophobic sidechains thereby rectifying an existing defect within the coiled-coil structure. Interestingly, the second coiled-coil substitution S477V (Fig. 5G) instead of forming an interaction along the 3-fold axis of the coiled-coil, establishes three hydrophobic contacts between the helical pairs, leaving a small cavity at the center.

### Stabilization strategy is applicable to other paramyxoviruses

Since paramyxovirus fusion proteins have similar architecture and fusion mechanisms, the local regions of instability that allow the structural transformation are likely also conserved. Therefore, we

investigated the possibility to transfer some of the stabilization solutions that were successful for RV3 preF to other paramyxovirus F proteins. The low structural complexity of the HRB stem region makes this an obvious region to investigate. Alignment of stem regions of representative paramyxovirus fusion proteins showed conserved imperfections in two *d* positions of the heptad repeats equivalent to residues 470 and 477 in RV3 of the HRB stem (Fig. 6A). Therefore, substitution to branched hydrophobic residues at the stem interface corresponding to the suboptimal *d* positions might prove a universal approach to stabilize the stem of paramyxovirus preF proteins. To test this hypothesis, transfer of stabilizing substitutions was attempted to RV1 F and NiV F, which share 42% and 24% sequence identity with RV3 F, respectively (Fig. 6B). For RV1 F, S473V and A480V substitutions were introduced separately or together in a non-stabilized RV1 F ectodomain lacking a heterologous trimerization domain. Prefusion F was detected in supernatant by BLI using preF-specific antibody 3 × 1[47]. Combined introduction of the HRB substitutions led to a 30% increase in preF binding (Fig. 6C), but the trimer remained below the detection limit of analytical SEC (data not shown). To improve expression of this RV1 construct, other RV3 preF-stabilizing substitutions were introduced as well (Fig. 6D). BLI binding rate analysis showed that a substitution equivalent to S41P in RV3 F (A44P) and the introduction of prolines in the β4-β5 hairpin of HRA increased RV1 F expression (Fig. 6D, Supplementary Fig. 7A, B). RV1 F that contained five stabilizing substitutions (A44P, E170P, Q171P, S473V and A480V) showed a high binding rate to antibody 3 × 1 (Fig. 6D) and eluted at a retention time consistent with a trimer on analytical SEC (Fig. 6E). Purification followed by SEC-MALS analysis of this stabilized RV1 preF confirmed the trimeric quaternary structure with an average Mw of 142 kDa (Fig. 6F). To further improve upon this construct, the equivalent of F335P from RV3, K339P, was introduced and tested in supernatant by analytical SEC, BLI and DSF. Introduction of the K339P substitution led to a further improvement in trimer expression and a 1.5 °C increase in thermal stability (Supplementary Fig. 7A–C).

NiV preF has been previously stabilized using a combination of a disulfide bond (L104C-I114C), a cavity-filling substitution (L172F), a proline in the HRA hinge (S191P), and a GCN4 trimerization domain[48]. To assess the feasibility of RV3-derived stabilizing substitutions in NiV preF, and to generate a stable NiV preF trimer devoid of heterologous trimerization domains, variants based on our stem-stabilization approach were made in a wildtype NiV F ectodomain without a GCN4 trimerization domain. The non-stabilized backbone hardly expressed as demonstrated by Octet with preF-specific antibody 5B3[49] and analytical SEC (Fig. 6G, H). Individual introduction of S470V or A477V did not lead to an observable increase in either BLI or SEC signals. However, when S470V and A477V were added in combination, 5B3 binding increased (Fig. 6G) and a peak was observed in analytical SEC at around 4.3 min that corresponds to a trimer (Fig. 6H). To further improve expression of this S470V + A477V variant, additional RV3 preF-based stabilizing substitutions were added. Proline substitutions in the β4-β5 hairpin of HRA increased preF expression levels of NiV F as shown by both binding to antibody 5B3 in BLI (Fig. 6I) and analytical SEC (Fig. 6J). Introduction of proline substitutions equivalent to RV3 F's S41P in the long connecting strand between DI and DIII (K49P) reduced preF expression as measured by BLI (Fig. 6I), and no trimer could be detected in analytical SEC (Fig. 6J). Expression of the variant with the conservative A165P substitution in combination with S470V and A477V gave among the highest expression levels according to both BLI and analytical SEC (Fig. 6I, J), but was still considered relatively low with a peak height in analytical SEC of ~1 mAU. To further improve its expression, stabilizing salt bridges were introduced at *b*, *c* and *g* positions of HRB based on the sequence of RV3 F (Fig. 6A) and tested in both BLI (Fig. 6K) and analytical SEC (Fig. 6L). Four out of 8 substitutions –S466D, Q469E, Y473E, L480K– increased preF expression dramatically, with Y473E improving preF expression 10-fold. This design with

greatly improved expression allowed us to retest the K49P substitution in the DI-DIII strand. Similarly, in this backbone the substitution reduced preF expression levels as measured by analytical SEC (Supplementary Fig. 7D) and BLI (Supplementary Fig. 7E), however, sufficient preF protein levels remained to perform heat-SEC (Supplementary Fig. 7D) which showed a strong increase of preF stability of the K49P variant of at least 8 °C. The equivalent to F335P from RV3 preF was also tested in the backbone with Y473E, and also showed a substantial increase in preF stability as measured by heat-SEC (Supplementary Fig. 7D), despite a small reduction in expression (Supplementary Fig. 7D, E).

In conclusion, both HRB stem-stabilizing substitutions, the prolines in the β4-β5 hairpin of HRA, the S41P in the DI-DIII β-strand and the F335P in the β-hairpin are transferable to divergent paramyxoviruses. Moreover, as illustrated by the Y473E substitution in NiV F, the HRB stem region emerges as a general region of interest for improving preF trimer expression.

## Stabilized preF but not postF induces neutralizing antibodies and provides protection against lower respiratory tract infection

To evaluate the RV3 preF protein OnlyEcto as a potential vaccine candidate, the induction of neutralizing antibody titers was compared between OnlyEcto and postF in naïve mice. Mice were immunized intramuscularly at weeks 0 and 4 with 15 µg of RV3 OnlyEcto or postF protein adjuvanted with 100 µg of alum adjuvant (Adju-Phos®). All mice generated preF and postF binding antibody titers above background of the mock control. As expected, both immunogens elicited significantly higher antibody binding titers to the homologous protein (Fig. 7A, B). Next, the functionality of the elicited antibodies was tested in a virus neutralization assay (VNA) using the JS strain of RV3 expressing GFP upon infection. Whereas neutralizing antibody titers induced by postF were not substantially above background levels observed in mock immunized mice, the OnlyEcto protein induced 18.9-fold higher VNA titers compared to the postF protein in pooled serum samples (Fig. 7C). The same serum pools of OnlyEcto and postF immunized mice were also tested in a VNA using differentiated primary human airway epithelial cells (hAEC) in combination with the RV3 JS-GFP strain. These hAEC cultures recapitulate human lung physiology and are considered an accurate model to study virus neutralization[50]. The postF serum pool did not show any neutralization whereas the OnlyEcto serum pool did show neutralizing activity, with viral breakthrough only observed at the 1:400 diluted serum (Fig. 7D).

To further characterize the RV3 preF immune response, naive mice were immunized at weeks 0 and 4 with 1.5, 5 or 15 µg OnlyEcto without adjuvant. OnlyEcto induced RV3 preF binding and neutralizing antibody titers in a dose-dependent manner as measured two weeks post 2nd dose (Fig. 7E, F).

Since adult humans have pre-existing immunity to RV3, we also assessed the ability of the stabilized preF protein to boost RV3 pre-existing immune responses in a pre-exposure mouse model. To this end, mice that had previously been intranasally infected with RV3 were immunized 19.5 weeks later with 1.5 µg OnlyEcto without adjuvant. Six weeks after immunization, preF binding and RV3 neutralizing antibody titers were measured. The preF protein was able to boost the pre-existing RV3 immunity as evidenced by significantly higher levels of preF binding (173.1-fold) and RV3 neutralizing (6.7-fold) antibody titers in the RV3 pre-exposed and immunized mice as compared to the mice that were only pre-exposed (Fig. 7G, H).

The ability of OnlyEcto to confer protection against RV3 was assessed in a RV3 cotton rat challenge model. Cotton rats were immunized twice with a 4-week interval with OnlyEcto (either unadjuvanted, or adjuvanted with AlOH (Janssen) or AS01$_B$) or with postF adjuvanted with AS01$_B$. Additionally, a control group was intranasally

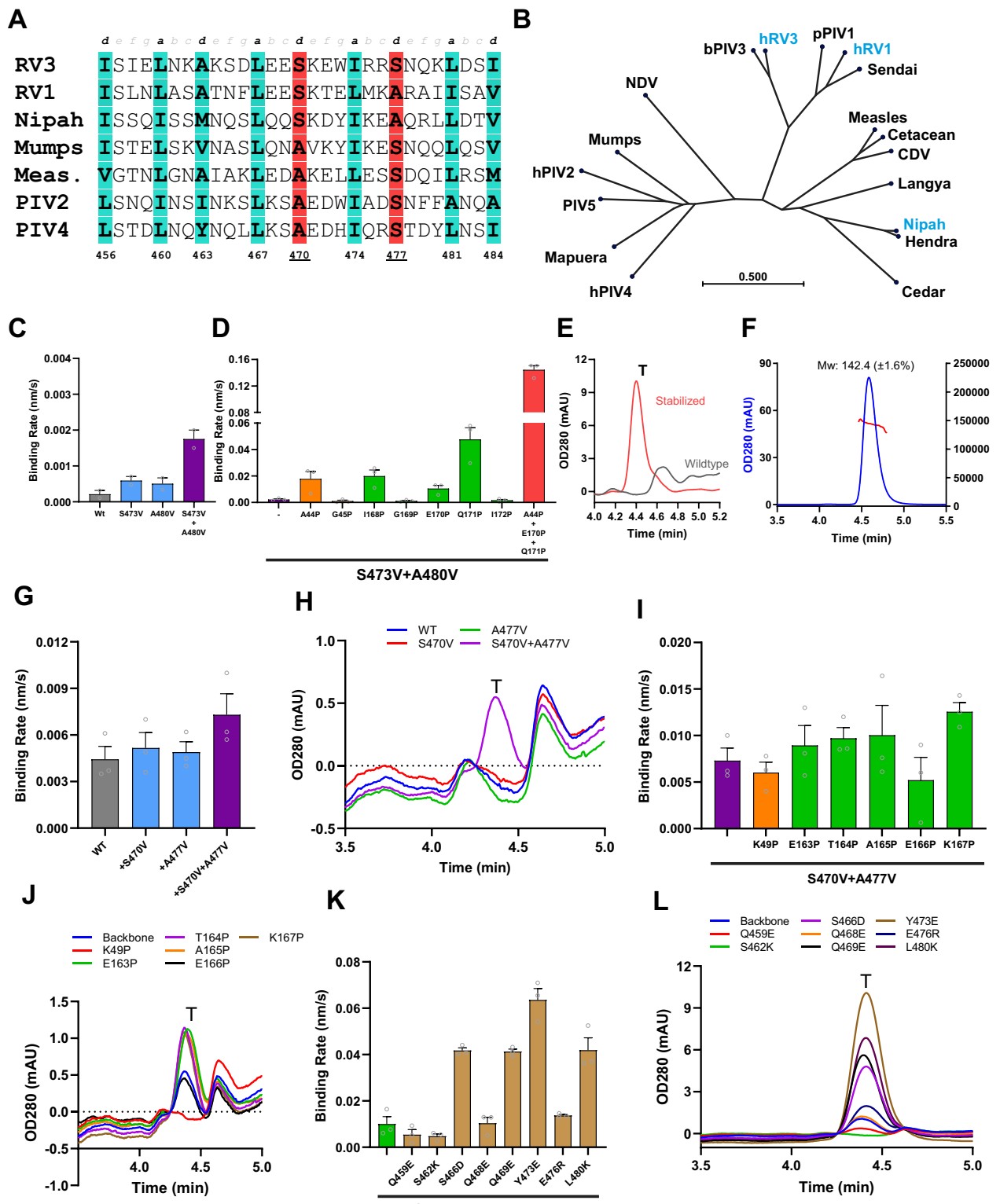

**Fig. 6 | Transfer of stabilizing substitutions to other paramyxovirus F proteins.** **A** Sequence alignment of the HRB heptad register of selected F proteins. The *a* and *d* positions are highlighted in cyan, with the suboptimal residues at positions 470 and 477 highlighted in red (numbering according to RV3). **B** Neighbor joining phylogenetic tree of representative paramyxovirus F proteins. **C**, **D** Expression of RV1 F protein variants in supernatant as measured by BLI using immobilized 3 × 1 antibody. The initial slope, V0, at the start of binding is plotted as the average of three independent transfections; error bars represent the SD. RV1 F substitutions A44P, E170P, Q171P, S473V, and A480V substitutions are the equivalents of the

following substitutions in RV3, respectively: S41P, N167P, L168P, S470V, and S477V. **E** Analytical SEC trace of wildtype and stabilized (A44P + E170P + Q171P + S473V + A480V) RV1 F in cell supernatant. The trimer (T) peak is indicated. **F** Analytical SEC-MALS of purified stabilized RV1 preF. Expression of NiV F protein variants in supernatant as measured by BLI using immobilized 5B3 antibody (**G**, **I**, **K**), or by analytical SEC (**H**, **J**, **L**). For BLI, the initial slope, V0, at the start of binding is plotted as the average of three independent transfections; error bars represent the SD. NiV F substitutions K49P, K167P, S470V, and A477V are the equivalents of the following substitutions in RV3, respectively: S41P, N167P, S470V, and S477V.

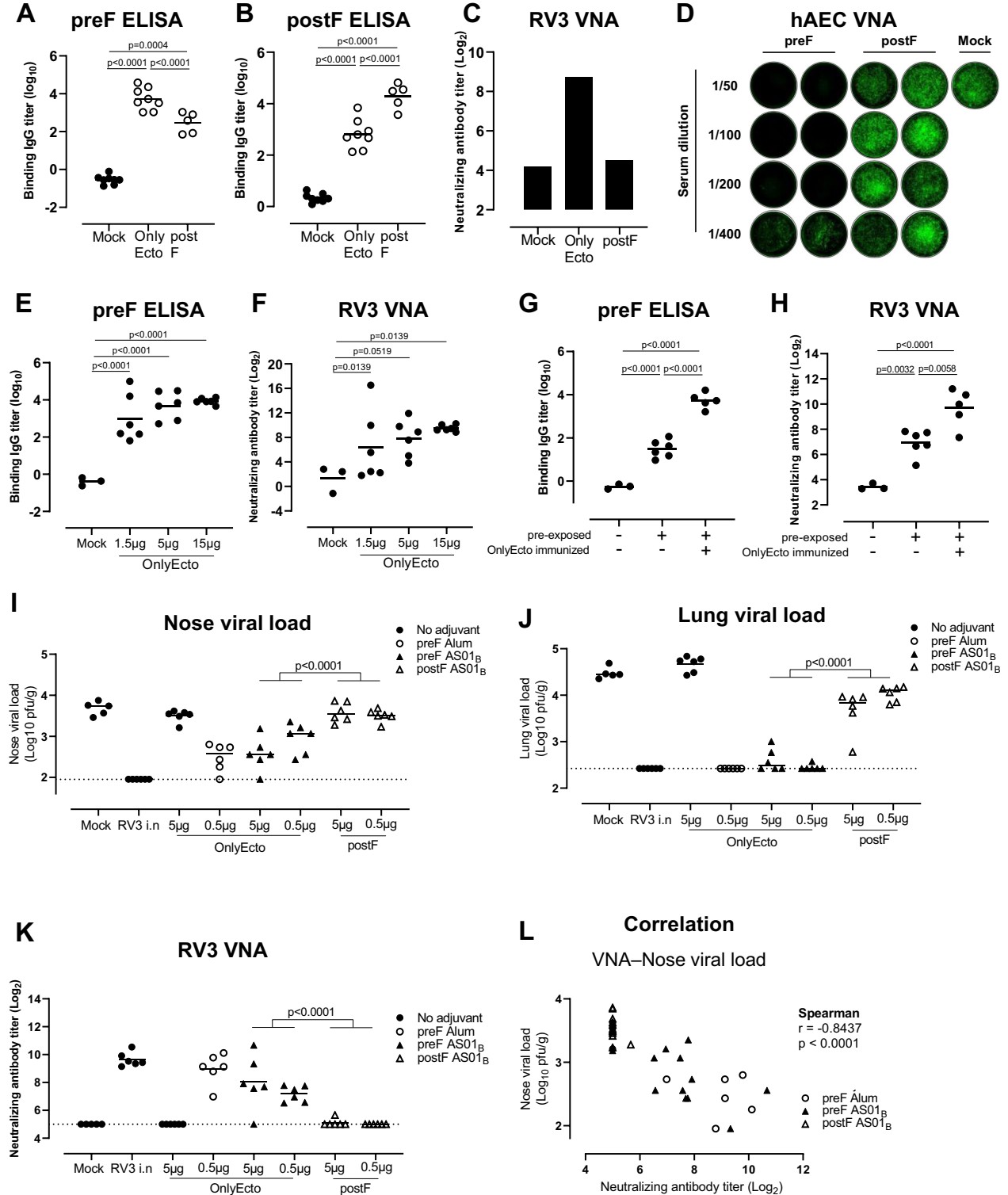

exposed to live RV3. Three weeks after the last immunization, cotton rats were intranasally challenged with RV3. Lung and nose viral loads were determined 4 days post challenge. Mock immunized animals developed robust nose and lung viral loads while animals that were intranasally pre-exposed with RV3 prior to challenge were completely protected against infection as evidenced by an undetectable viral load in lung and nose tissue samples. Viral titers in animals immunized with OnlyEcto without an adjuvant were not different from the mock immunized control animals. In contrast, animals immunized with adjuvanted OnlyEcto showed near complete protection in the lower

respiratory tract, and significantly lower viral load in the nose. Animals immunized with AS01$_B$ adjuvanted postF had significantly higher viral loads in the lungs as compared with animals immunized with OnlyEcto in an across dose level comparison (Fig. 7I, J). Virus neutralizing titers were measured in pre-challenge serum. In contrast to the immunogenicity data observed in mice, OnlyEcto needed an adjuvant to be immunogenic in cotton rats. OnlyEcto induced RV3 neutralizing antibody responses when adjuvanted with AlOH or AS01$_B$ while adjuvanted postF was unable to do so. (Fig. 7K). RV3 neutralizing antibody titers correlated with nose viral loads in animals that were immunized

**Fig. 7 | Preclinical evaluation of RV3 preF (OnlyEcto). A–D** Mice (*n* = 5 or 8) were immunized with 15 μg adjuvanted preF or postF protein or formulation buffer (Mock) at week 0 and 4. Two weeks later, serum samples were taken and preF (**A**) and postF (**B**) binding antibody titers were measured by ELISA or virus neutralization titers (VNT) in preF, postF or Mock pooled serum samples were determined with an VNA (virus neutralization assay) on Vero cells (**C**) or on differentiated human airway epithelial cell cultures (**D**) using the RV3 JS strain equipped with a GFP reporter gene. **E, F** Mice (*n* = 5 or 8) were immunized with a dose range of 1.5, 5 or 15 μg non-adjuvanted OnlyEcto or formulation buffer (Mock) at week 0 and 4. Two weeks later serum samples were taken and preF binding antibody titers were measured by ELISA (**E**) or RV3 neutralizing antibody titers by RV3-GFP VNA on Vero cells were determined (**F**). **G, H** Mice (*n* = 3 or 6) were intranasally exposed to RV3 or formulation buffer as control and immunized 19 weeks later with OnlyEcto or formulation buffer. Six weeks later, serum samples were taken and preF binding

antibody titers were measured by ELISA (**G**) or RV3 neutralizing antibody titers by RV3-GFP VNA on Vero cells were determined (**H**). Analysis of variance (ANOVA; 2-sided t-test) was used for statistical comparisons between groups. Tukey-Kramer (**A**, B and **G**, **H**) or Dunnett adjustments. **E**, **F** for multiple comparisons were applied. **I–L** Cotton rats (*n* = 5 or *n* = 6) were immunized with OnlyEcto with or without adjuvant or postF with adjuvant or formulation buffer (Mock) at week 0 and 4. Three weeks after the final immunization, cotton rats were challenged intranasally with RV3. Four days later viral loads in nose and lung tissue were determined (**I**, **J**). RV3 neutralizing antibody titers by plaque reduction neutralization test (PRNT) were determined in pre-challenge sera (**K**) and the correlation between nose viral loads and VNA titers were calculated with a spearman correlation analysis (**L**). Mean responses per group are indicated with horizontal lines. Statistical comparisons were performed across dose levels using a Tobit model (**I–L**). *P* < 0.05 were considered statistically significant.

with OnlyEcto or postF (Fig. 7L), suggesting an important role for neutralizing antibody responses in mediating protection.

## Discussion

Recent insights have revealed that stabilizing the prefusion conformation of viral fusion proteins can significantly enhance vaccine efficacy, because more neutralizing epitopes are present on preF than on postF[36,37,39,51]. This principle proved crucial for generating efficacious RSV and SARS-CoV-2 vaccines and holds promise for other respiratory viruses, such as RV3. Similarly to RSV F, the wildtype, non-stabilized RV3 F ectodomain fails to generate detectable levels of the trimeric prefusion form of the protein, making it unsuitable as a vaccine candidate. In this study, we engineered a protein vaccine candidate by introducing substitutions that stabilize the prefusion state, resulting in a preF protein design that expresses to high yield, exhibits excellent thermal stability and elicits potent neutralizing antibodies in vivo.

The stabilizing substitutions implemented in our vaccine candidate can be classified into two categories. Firstly, we repurposed the inherently unstable coiled-coil stem, encompassing HRB, as a natural trimerization domain. Substitution of the buried hydrophilic Ser residues located at positions 470 and 477 with Val residues resulted in a strong increase of trimer formation and stability without the requirement for a heterologous GCN4 trimerization domain. The immunogen will therefore not induce or boost nonrelevant GCN4 antibodies upon repeated vaccination or vaccination with possible future GCN4-containing vaccines. Secondly, we stabilized the head domain through the introduction of four substitutions. The L168P substitution in the β4-β5 hairpin loop of HRA impedes α-helix formation during transition into the postfusion state and is reminiscent of the stabilizing 162C-168C disulfide described previously[30]. Although both N167P and L168P substantially improved preF trimer expression, only L168P completely prevented the formation of syncytia. This can likely be attributed to the positioning of L168 at the critical *d* position in the postfusion coiled-coil conformation.

The other head-stabilizing substitutions are located within the same horizontal plane of the prefusion head (Supplementary Fig. 8) and stabilize directly or indirectly the dynamic HRA as well as the relatively constant 'DIII core' composed of an outer layer that includes the HRC helix of F2 (residues 64-94), the long β-sheet of DIII that connects to DI, and an inner layer that includes the helical bundle region (HB), an assembly of helices between residues 216 and 266 (Fig. 5A). The F335P substitution (Fig. 5E), in the beta hairpin adjacent to the S41 hydrophobic pocket, both eliminates the unusually solvent-exposed hydrophobic F335 while simultaneously reinforcing the turn structure and forming a favorable interaction with Y254 which allows interactions with the adjacent protomer (Fig. 5A). Substitution of the buried hydrophilic for hydrophobic residues in the DIII core (Q89M and Q222I) creates a cluster of contacts between three helical regions (HRA-HRC-HB) including the

217–224 helix in HB, which was disordered or random coil in previous structures[30,47], and the fusion peptide proximal HRA helix. Stabilization of this cluster impedes refolding of HRA and locks the fusion peptide in place thereby reinforcing the preF conformation. The stabilizing S41P substitution also restricts release of the fusion peptide. Located at the bend in the long ß-sheet that runs between DI and DIII it occupies an intra-protomeric hydrophobic pocket formed by a cluster of four sequence-distant groups of residues. Ser41 has both unfavorable Phi and Psi-angles as well an unfavorable positioning of its hydrophilic sidechain in this hydrophobic environment. Moreover, during the transition from the preF to the postF conformation, the twisted ß-strands in the long DI-DIII ß-sheet rearrange due to improved H-bonding. As a result, domains DII and DI rotate inward to the trimer axis (Fig. 8A, B), leading to the release of the fusion peptide that was wedged between adjacent protomers (Fig. 8C). S41P impedes this motion due to the permissible Ramachandran angles for proline as well as the favorable interactions of the proline side chain in the hydrophobic pocket.

Although HRA and HRB show the most dramatic conformational changes during the fusion steps, the long DI-DIII ß-sheet and especially the helical bundle in the DIII core also show structural reorganization (Fig. 8D, E). The helical bundle connects most of the stabilized regions since it is part of the HRA-HRC-HB cluster, the 41P hydrophobic pocket and connects to the HRA-HRC-HB cluster of the adjacent protomer (Fig. 5A, Supplementary Fig. 8). The marked reorganization of the helical bundle and the DI-DIII ß-sheet during the preF to postF transformation results in loss of all these contacts. These structural changes and the concomitant movement and rotation of DII and DI are not observed for the related pneumovirus F (Fig. 8E) and is a specific feature of paramyxovirus fusion protein conformational change that is needed to release the fusion peptide and fusion peptide proximal region.

Paramyxoviruses exhibit notable host and sequence diversity, signifying their considerable zoonotic potential and the likelihood of interspecies transmission[52,53]. To prepare for a potential pandemic caused by an emerging virus, rapid development of medical countermeasures, including efficacious vaccines, is crucial. This was also highlighted by the COVID pandemic, where previous research into RSV F stabilization and coronavirus Spike stabilization facilitated the design of the prefusion-stabilized immunogen[36,39]. Here, we showed that stabilization of the regions involved in structural transformation of paramyxovirus fusion proteins can be transferred to other paramyxovirus F proteins, with most RV3 substitutions also being successful in closely related RV1 F, and even in the more distant NiV F protein. It will be interesting to ascertain whether stabilization of the HRB coiled-coil, which is a more universal structural element of class I fusion proteins, is transferable to pneumovirus F and coronavirus Spike.

In the RSV field, it is widely acknowledged that the prefusion conformation of the F protein is required to induce strongly neutralizing antibodies[37]. Here, in line with previous preF versus postF

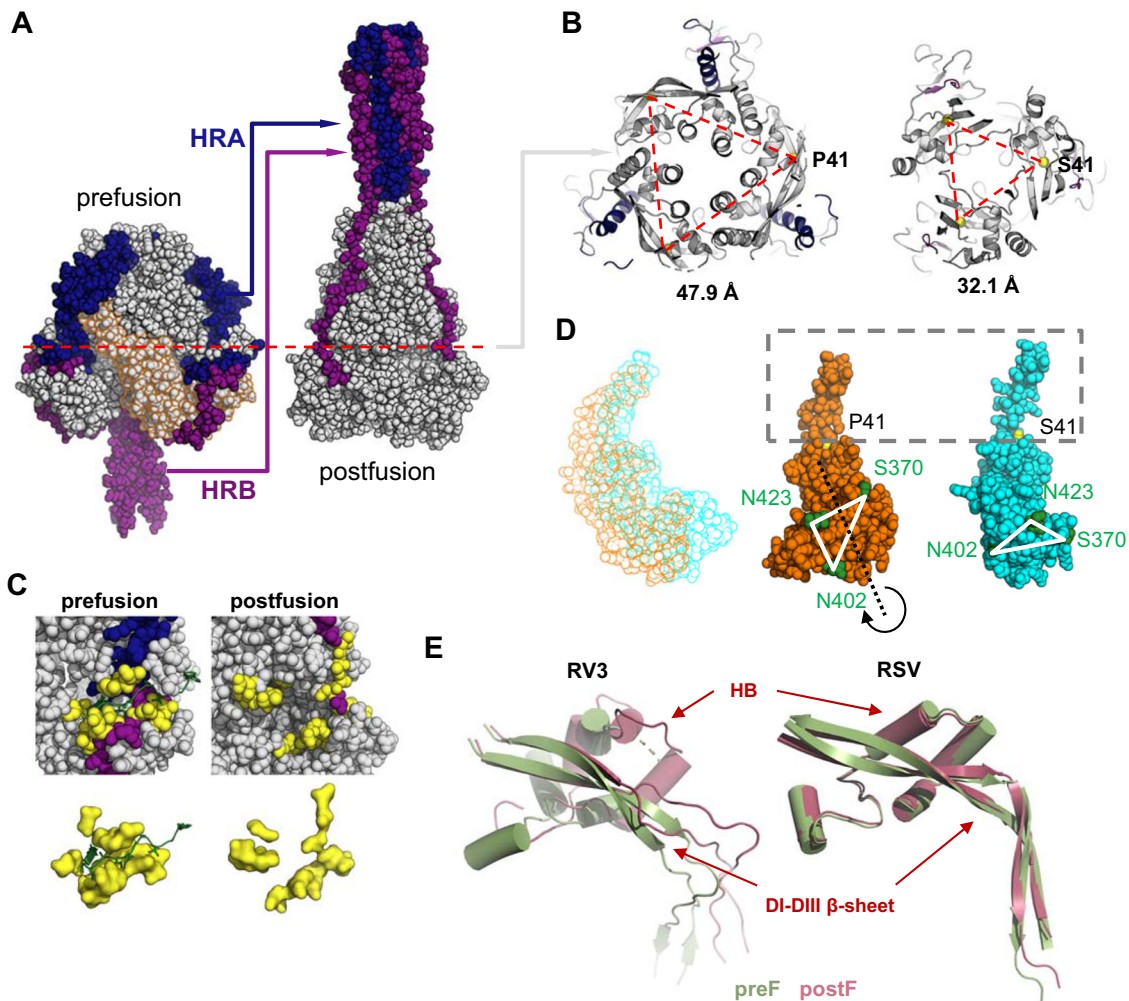

**Fig. 8 | Structural transformation of the RV3 preF conformation. A** The prefusion structure (OnlyEcto2P) compared with the postfusion structure of RV3 F (PDB ID 1ZTM[65]). HRA and HRB, including the preceding loop, are colored in blue and purple respectively. DI and DII in the prefusion structure are indicated with an orange outline. **B** Cross-section indicated in (**A**) for the prefusion (left) and postfusion (right) structures. C-alpha of residue 41 has been indicated as yellow sphere at the corner of the triangle and the distances between the C-alpha's of the three monomers are indicated below. **C** Fusion peptide (green) pocket (yellow) for the preF and postF structures were plotted in the context of the full protein (upper panels) and without it (lower panels). **D** *left panel* Translation of DI and DII during the conformational change, with the preF structure indicated in orange and postF in cyan. *Middle and right panel* Aligned on the beta sheet around residue 41, with three residues in green to indicate the rotation of the domain during the refolding. **E** Comparison of the conformational change of the DI-DIII β-sheet structure and helical bundle (HB) between RV3 F and RSV F, with the preF structure indicated in green and postF in red. The structures were aligned based on the top part of the DI-DIII β-sheet.

comparisons for both RSV and RV3[30,35], we show that both preF and postF protein are highly immunogenic in mice, but that only preF induces potent neutralizing responses.

In contrast to the mouse immunogenicity data, preF antigen needed an adjuvant to be immunogenic in the cotton rats. When adjuvanted, preF induced superior protection against RV3 infection compared to postF, with almost complete protection from infection in the lower respiratory tract and partial protection in the upper tract. This suggests that the preF vaccine candidate applied intramuscularly is able to protect against lower tract infection but could be less effective against viral spreading from the upper respiratory tract. There was a strong correlation between neutralizing antibody titers and protection against RV3 challenge, suggesting an important role for neutralizing antibody responses in mediating protection. However, it is known that there is also a role for CD8 + T cells in protection against respiratory viruses[54], and we cannot exclude that these play a role in the observed protection. To identify the immunological mechanism of protection mediated by RV3 preF based vaccines

further characterization of the antibody and cellular immune responses and their interplay are required.

Most people have experienced repeated exposure to RV3[55,56] implying that a vaccine will likely recall pre-existing immunity. We could demonstrate recall of neutralizing antibody titers in a RV3 pre-exposure model showcasing that an RV3 preF is an attractive candidate vaccine antigen.

Novel RSV preF protein-based vaccines have shown to be highly effective in preventing RSV disease in several phase 2b and 3 clinical studies and will pave the way for other subunit-based vaccines against viral pathogens causing respiratory disease in humans[37,51]. By utilizing purified subunit protein antigens, these vaccines offer precise antigenic characterization, leading to optimal coverage and reduced risks compared to whole-virus, live attenuated, and vector-delivered vaccines. As a result, they represent a promising approach to combat respiratory viral diseases and warrant further investigation to fully harness their potential and be better prepared for potential future pandemics.

## Methods

### DNA synthesis

DNA fragments encoding the different RV3 F proteins based on the HPIV3/Seattle/USA/10Q7/2010 strain (Genbank ID: ARQ32975.1) were codon-optimized, synthesized and cloned into a pCDNA2004 vector at Genscript. Endotoxin free stocks were prepared and used for transfection experiments. For generation of RV3 postF protein used in preclinical experiments a furin site ('RRRR') was introduced at the F2/F1 boundary and the first 11 residues of the fusion peptide were deleted[30].

### 96-well transfection

For screening experiments Expi293F cells (Thermo Fisher; A14527) were transfected at a 200 µL scale at a viable cell (vc) density of 2.5E + 06 vc/mL using the ExpiFectamine 293 Transfection Kit (Gibco, Thermo Fisher Scientific) according to the manufacturer's instructions in Expi293F Expression medium [+] GlutaMAX (Gibco, Thermo Fisher Scientific) with 100 U/ml Pen-Strep (Gibco, Cat# 15140122) in 96-half deep-well plates. The cells were cultured for three days at 37 °C, 75% humidity, 250 rpm, 8.0% $CO_2$ after which the supernatants were harvested by centrifugation (10 min at 500 x $g$) and clarified by both a filtration step (96-well 350 µL AcroPrep Advance Supor Filter plates, 0.2 µm, Pall) and a centrifugation step (10 min at 500 x $g$). The obtained supernatant was stored at 4 °C until further use. Mock transfections without the addition of plasmid were used to perform baseline corrections in analytical SEC and BLI experiments.

### Large scale transfection

Supernatant for purification was obtained by transient transfection of Expi293F cells in 300 mL scale at a cell density of 2.5E + 06 vc/mL using the ExpiFectamine 293 Transfection Kit (Gibco, Thermo Fisher Scientific) according to the manufacturer's instructions in Expi293F Expression medium [+] GlutaMAX (Gibco, Thermo Fisher Scientific) in 1 L vented flasks (Corning). One day post transfection enhancers were added. Five days post transfection and culturing at 37 °C, 75% humidity, 125 rpm, 8.0% $CO_2$ the supernatants were harvested by centrifugation (10 min at 600 x $g$) and clarified by sterile filtration using a 0.22 µm PES vacuum filter (Nalgene). The obtained supernatant was stored at 4 °C until further use.

### Biolayer interferometry using monoclonal antibodies

Antibodies at a concentration of 5 µg/ml in 1× kinetics buffer (Sartorius, cat. #18-1105) were immobilized to anti-hIgG (AHC) sensors (FortéBio, cat. #18–5060) in 96-well black flat-bottom polypropylene microplates (Corning, cat. #3694) during a 30 min association step. Mock supernatant and supernatant containing RV3 F protein were also in 96-well black flat-bottom polypropylene microplates. An Octet RED384 system (FortéBio) was used to perform the experiment at a shaking speed of 1000 rpm at 30 °C. Sensors were activated in 1x kinetic buffer for 600 s, followed by immobilization of antibody for 1800 s, blocking in mock supernatant for 600 s and binding of the RV3 protein for 900 s. Data analysis was performed using FortéBio Data Analysis 12.0 software (FortéBio) and the binding rate (V0) was determined over the complete association phase and represented as nm/s.

### Heat-SEC on cell supernatant

Per temperature point, 45 µL of supernatant containing RV3 protein was aliquoted in a 500 µL Eppendorf tube. The supernatant was heated at 50 °C or 60 °C (Eppendorf ThermoMixer C) for 15 min, 0 rpm, whereas the control was kept at 4 °C. Subsequently, the samples were centrifuged at 18,000 x g for 10 min. Read out was performed by applying 20 µL sample to a Unix-C SEC-300 15 cm column (Sepax Technologies) with the corresponding guard column (Sepax Technologies) at 25 °C equilibrated in running buffer

(150 mM sodium phosphate, 50 mM NaCl, pH 7.0) at 0.35 mL/min on a an ultra-high-performance liquid chromatography system (Vanquish, Thermo Scientific). Analytical SEC data was analyzed using Chromeleon 7.2.8.0.

### SEC-MALS on purified proteins

Purified protein was characterized by using an ultra-high-performance liquid chromatography system (Vanquish, Thermo Scientific) and µDAWN TREOS instrument (Wyatt) coupled to an Optilab µT-rEX Refractive Index Detector (Wyatt), in combination with an in-line Nanostar DLS reader (Wyatt). A maximum of 10 µg protein or 20 µL was applied to a Unix-C SEC-300 15 cm column (Sepax Technologies) with the corresponding guard column (Sepax Technologies) at 25 °C equilibrated in running buffer (150 mM sodium phosphate, 50 mM NaCl, pH 7.0) at 0.35 mL/min on a an ultra-high-performance liquid chromatography system (Vanquish, Thermo Scientific). The data was analyzed using Chromeleon 7.2.8.0 software package, and conformation, hydrodynamic radius and molecular weight of RV3 F trimers were calculated by Astra 8.0.0.19 software (Wyatt) using a dn/dc value of 0.185 for the protein component and 0.1410 for the glycan component. Molecular weights were calculated using the RI detector as source for concentration and mass recoveries using UV as source for concentration.

### DSF

PBS pH 7.4 (1x, Gibco) was added to 20 µg of purified protein to a total volume of 90 µl to which 10 µL 50x SYPRO Orange (5000× stock diluted in PBS, Invitrogen S6650) was added. A negative control containing no protein was included for potential reference subtraction. Triplo's of 30 µL were dispensed in a MicroAmp Fast Optical 96-well Plate (Thermo Fisher) and sealed with MicroAmp Optical Adhesive Film (Thermo Fisher). The samples were heated using a temperature ramp from 25 to 95 °C with a rate of 0.015 °C per second. Data was collected continuously by the qPCR instrument (Applied Biosystems ViiA 7) measuring reporter ROX. The melting temperatures were derived from the negative first derivative of the fluorescent signal, which was plotted as a function of temperature. The lowest point in the curve indicated the melting temperature $Tm_{50}$.

### Fusion assay

Cell-cell fusion assays were performed to determine the effect of stabilizing substitutions on F fusogenicity. To this end, 1E + 05 of Human Embryonic Kidney (HEK) 293 cells (ATCC; CRL-1573) per well were seeded in 500 µL of Dulbecco's Modified Eagle Medium (DMEM) without phenol red (Gibco, cat# 21063-029) and supplemented with 10% Fetal Bovine Serum (FBS, Gibco, cat# 10091-148) in 24-well plates the day before transfection. Cells were transfected the next day with 375 ng of pcDNA2004 plasmid encoding the full-length wildtype F protein sequence of the JS strain of RV3 or stabilized variants thereof, 25 ng of HN with H552Q mutation, and 100 ng of mScarlet per well using the TransIT-LT1 transfection reagent (Mirus, cat# MIR 2305) according to the manufacturer's protocol with a 2:1 ratio of TransIT-LT1 Reagent to DNA ratio. The cells were incubated overnight at 37 °C and 10% $CO_2$. To visualize the cell's nuclei, NucBlue live cell stain ReadyProbes (Hoechst 33342 based, Invitrogen, cat# R37605) was applied to the transfected cells 15 min before they were imaged at 10x magnification using an EVOS M5000 cell imaging system (Thermo Fisher Scientific). Composite images of the Red channel (mScarlet) and Blue channel (Hoechst) were made using the ImageJ software package[57]. The mScarlet fluorophore expressed in the transfected cells will color the cytosol in red, and the NucBlue stain is used to visualize the nuclei. When a transfected cell fuses with a non-transfected cell, the red signal becomes diluted due to merger of red and non-red cells. Moreover, due to gravity the nuclei tend to float towards each other in syncytia.

## C-tag purification

C-tagged RV3 F trimers were purified using a two-step protocol on an ÄKTA Avant 25 system (Cytiva). 0.22 μm filtered supernatant was applied to a 5 ml capture Select C-tag XL column (Thermo Scientific) equilibrated in PBS pH 7.4 (1x, Gibco) after which the column was washed with PBS and the protein was eluted with 20 mM Tris, 2 M MgCl2, pH 7.0 and 1:1 diluted with 20 mM Tris, pH 8.0 to lower the $MgCl_2$ concentration. The obtained protein was concentrated using an Amicon Ultra 15–30-kDa cutoff filter (Millipore) prior to further purification via size exclusion. The protein was applied on a HiLoad Superdex 200 16/600 column (Cytiva) equilibrated in 20 mM Tris, 75 mM NaCl, pH 7.5 and monodisperse fractions were pooled and filtered 0.22 μm to form the final product. Proteins could be stored at 4 °C or snap frozen in liquid nitrogen for long term storage at −80 °C.

## Tagless purification

RV3 F trimers that did not have an affinity tag were purified using a two-step protocol on an ÄKTA Avant 25 system (Cytiva). 0.22 μm filtered supernatant was diluted with 1.5 volume of Milli-Q water and conditioned to pH8 by adding 200 mM Tris, pH 8.0 to a final concentration of 40 mM Tris. The conditioned supernatant was applied to a 5-ml Capto Q Impress column (Cytiva) equilibrated in 20 mM Tris, pH 8.0. A mixture of 5% buffer B (20 mM Tris, 1 M NaCl, pH 8.0) in buffer A (20 mM Tris, pH 8.0) was used to wash the column, after which the protein was eluted stepwise with 15%, 30%, 50% and 100% buffer B with a length of 4 column volumes per step. RV3 F trimer that eluted in the 15% buffer B step was pooled and concentrated using an Amicon Ultra 15–30-kDa cutoff filter (Millipore) prior to further purification via size exclusion. The protein was applied on a HiLoad Superdex 200 16/600 column (Cytiva) equilibrated in 20 mM Tris, 75 mM NaCl, pH 7.5 and monodisperse fractions were pooled and filtered 0.22 μm to form the final product. Proteins could be stored at 4 °C or snap frozen in liquid nitrogen for long term storage at −80 °C.

## Cryo-EM sample preparation and data collection

Purified RV3 preF (OnlyEcto2P) at 1 mg/mL was combined with 1.25 molar excess VHH 4C06 in buffer containing 2 mM TRIS pH 8.0, 200 mM NaCl, 0.02% NaN₃, and 0.01% amphipol. After allowing the complex to incubate on ice for 60 min, 3 μL of sample was applied to a C-flat Protochip 1.2/1.3 grid (Electron Microscopy Sciences CF413-50) that had been plasma cleaned for 40 s using a 4:1 ratio of $O_2$:$H_2$ in a Solarus 950 plasma cleaner (Gatan). Excess liquid was blotted from the grid using a Vitrobot Mark IV (FEI) set to 100% humidity and 4 °C with an applied blot force of 0 for 3 s before plunge-freezing into liquid ethane.

A total of 2061 movies were collected from a single grid at a 30̊ tilt angle using a Titan Krios TEM (Thermo Fisher) equipped with a K3 detector (Gatan). Particles were imaged using SerialEM[58] with defocus values ranging from −1 to −2.5 μm at a calibrated magnification of 0.81 Å/pixel. Cryo-EM data collection parameters and refinement statistics are shown in Supplementary Table 1.

## Cryo-EM data processing and model building

Motion correction, CTF-estimation, blob particle picking, and particle extraction was performed for all movies in cryoSPARC Live v3.2.0[59]. Initial 2D classification and a single-class ab-initio reconstruction was determined for a subset of particles during data collection. Templates were created from 13 selected 2D classes containing a total of 201,729 particles. All subsequent processing was performed in cryoSPARC v3.2.0. Using the templates created in cryoSPARC Live, template-based particle picking was performed on 1690 accepted micrographs, yielding a total of 488,241 curated picks. After particle extraction, a four-class ab-initio reconstruction and subsequent heterogeneous refinement of all classes was performed. One class, constructed from 240,485 particles showed RV3 preF bound to 3 VHHs and was carried

forward to homogeneous refinement, followed by non-uniform refinement. To remove lower-quality particles from the particle stack and improve map quality, the high-resolution reconstruction from non-uniform refinement and two "junk" volumes were used to sort the remaining 240,485 particles using heterogeneous refinement. From this, 193,265 particles that were sorted into the target class were then used to perform two concurrent rounds of non-uniform refinement, yielding a 3.3 Å resolution reconstruction with no symmetry applied, and a 3.1 Å resolution reconstruction with applied C3 symmetry. The final C3 map was further processed using the DeepEMhancer tool within Cosmic[2] Gateway[60].

An initial model of RV3 preF was created by rigid-body fitting the RV3 preF cryo-EM structure (PDB ID: 6MJZ) into the 3.1 Å resolution C3 cryo-EM map using ChimeraX[61]. The complete model was built and refined iteratively using Phenix[62], Coot[63], and ISOLDE[64].

## Ethical statement

Mouse studies were conducted at Janssen Vaccines and Prevention B.V. according to the Dutch Animal Experimentation Act, and the Guidelines on the Protection of Animals for scientific purposes by the Council of the European Committee after approval by the Centrale Commissie Dierproeven and the Dier Experimenten Commissie of Janssen Vaccines and Prevention B.V.

Cotton rat study was conducted at Sigmovir Biosystems, Inc. by permission of the Institutional Animal Care and Use Committee (IACUC) of Sigmovir Biosystems, Inc.

## Animal experiments

BALB/c mice (Charles River Laboratories; female, 8 weeks) were immunized at week 0 and 4 via the intramuscular route with preF (OnlyEcto) or postF proteins with or without 100 μg Alum adjuvant per animal (Adju-Phos®; Invivogen) or formulation buffer or were intranasally exposed to $10^5$ pfu RV3 virus (VR-93™; ATCC) and intramuscularly immunized 19 weeks later with preF protein. Blood was collected via cardiac puncture at week 6 after the first immunization, after which the animals were sacrificed by cervical dislocation. Female cotton rats (Sigmovir Biosystems, Inc., Rockville, MD, USA; 6–8 weeks old) were immunized intramuscularly at week 0 and week 4 with indicated doses of OnlyEcto (preF), with OnlyEcto and postF protein adjuvanted with 50 μl AS01$_B$ (a component of Shingrix, GlaxoSmithKline, London, UK) or with OnlyEcto adjuvanted with 100 μg Alum adjuvant (AlOH, Janssen) or were intranasally exposed to live RV3 ($10^4$ pfu/animal, VR-93™; ATCC) at day 0 after which animals were challenged intranasally at week 7 with $1 \times 10^5$ pfu per animal of RV3 virus (VR-93™; ATCC). Animals were sacrificed 4 days post-challenge and viral load was determined by plaque assay in lung and nose tissue and expressed as pfu/g of tissue. Serum samples were collected prior to challenge via the retro-orbital route for readout of humoral immune responses.

Mice and cotton rats were housed in individual ventilated Green Line type II (501 $cm^2$) or type III (904 $cm^2$) (TECNIPLAST S.p.A., Italy) cages under controlled environmental conditions with a temperature range of 20 to 24 °C and a humidity of 55% -/+10%. A day and night light cycle (12h/12h) was maintained. All animal studies were conducted under class 2 biosafety level.

## PreF and postF binding antibody titers measured by ELISA

IgG antibodies binding to RV3 preF and postF proteins were measured by ELISA using white 96-well plates which were coated overnight at 4 °C with Streptavidin (0.66 μg/ml) for the preF ELISA or with postF protein for the postF ELISA. After washing, the wells were blocked with 2% bovine serum albumin (BSA) for 30 min at RT. In case of the preF ELISA, the plates were washed again, followed by addition of C-terminally biotinylated RV3 preF protein and incubation for 1 h at RT. After washing, serially diluted serum was added

and incubated for 1 h at RT. RV3 preF and postF binding antibodies were detected by HRP-labelled goat anti-mouse IgG (Biorad #172-1011; 1 h at RT) and after washing the wells were developed with LumiGlo (50 μl/well). The luminescence signal was measured with the Biotek Synergy Neo plate reader (Luminescence, filter 11: excitation 330 and filter 42: emission 495/520), and the relative potency was calculated to a standard serum sample taken along on each plate and expressed on a log10 scale.

### RV3 neutralizing antibody titers GFP-based VNA on VERO cells

VNA titers for RV3 were determined by a neutralization assay using RV3-susceptible Vero cells (received from the WHO; WHO10-87; 880101) and a recombinant RV3 virus (JS strain) that expresses GFP (RV3-GFP). The RV3-GFP virus was obtained from ViraTree (cat. P323; passaged 2 times) and propagated on LLC-MK2 cells (obtained from ATCC; CCL-7™) and used at passage 3. All sera were heat-inactivated at 56 °C for 30 min after which serially diluted sera were mixed with 2500 plaque-forming units (pfu) of RV3-GFP in 96-well black and white flat bottom tissue culture plates and incubated for one hour at RT. Subsequently $1 \times 10^4$ Vero cells per well were added and plates were incubated for 4 days at 37 °C, 10% $CO_2$. The monolayers were washed after which the GFP signal was determined with the Biotek Synergy Neo plate reader (Fluorescence filter set 107: excitation 485/20, emission 528/20). PIA174 at a concentration of 10 μg/ml, 1 μg/ml and 0.25 μg/ml was taken along as QC samples. Virus neutralizing titers (VNT) were calculated as the serum dilution that caused a 50% reduction in GFP signal and expressed as log2 IC50 titer.

### RV3 neutralizing antibody titers using differentiated human airway cell (hAEC) cultures

Fully differentiated human airway epithelial cells (hAEC) of nasal origin (MucilAir) and grown on an air-liquid interface were purchased from Epithelix Sarl (Geneva, Switzerland). The inserts consisted of cells from a pool of 14 anonymized donors. The RV3-neutralizing capacity of pooled mouse sera was tested in hAEC's infected by RV3-GFP (JS strain), which was purchased from ViraTree (North Carolina, USA). Ready-to-use MucilAir inserts were maintained at an air-liquid interface according to the manufacturer's instructions for 4 days prior to the start of the experiment. Each hAEC insert had undergone prior testing by the manufacturer to ensure ciliary beating, polarization of the epithelial layer and mucus production, as corresponding to healthy respiratory epithelium. At the start of the experiment, inserts were washed once with 200 μL PBS to remove mucus and cell debris. RV3-GFP ($5 \times 10^6$ TCID50 units) diluted to a final volume of 25 μL 1xPBS, was gently mixed 1:1 with 25 μL diluted serum, before being added to the apical side of the inserts. After a 1 h incubation at 37 °C, the serum/virus mixture was aspirated. Negative controls were infected in presence of a 50-fold dilution of serum from mock-vaccinated mice. Inserts were incubated for 96 h at 37 °C post-infection before the GFP fluorescent signal was visualized using a BioTek Cytation 1 automated microscope (Agilent) using a 2.5x plan achromat objective (Meiji). The individual images were stitched using the Gen5i Plus v3.08.01 software package.

### RV3 neutralizing antibody titers using Plaque reduction neutralization test

The cotton rat pre-challenge serum samples were analyzed for RV-3 neutralizing antibody titers by Plaque reduction neutralization test (PRNT). Heat inactivated serum samples were serially diluted and incubated with 25–50 pfu RV3 (strain VR93) for 1 h at room temperature and inoculated in duplicates onto confluent LLC-MK2 monolayers. After one hour incubation, the wells were overlayed with Methylcellulose medium. After 4 days of incubation, the overlay was removed, and the cells were fixed with 0.1% crystal violet stain for one hour and then rinsed and air dried. Plaques were manually counted, and titers were calculated as the proportion pfu's over total (pfu count in the reference wells) by Probit regression to estimate the serum dilution corresponding to a 50% reduction in plaque counts.

### Statistical analysis

For the ELISA and VNA data an analysis of variance (ANOVA) was used for statistical comparisons between groups. The data were log2 or log10 transformed (VNA, ELISA, respectively). Tukey-Kramer or Dunnett adjustments for multiple comparisons were applied. For the cotton rat challenge study, the lung and nose viral loads and PRNT titers of the treatment groups were compared with the mock control group and OnlyEcto and postF adjuvanted with AS01$_B$ were compared across dose levels using a Tobit model. Statistical analyses were performed using SAS version 9.4 (SAS Institute, Inc., Cary, NC, USA).

### Reporting summary

Further information on research design is available in the Nature Portfolio Reporting Summary linked to this article.

## Data availability

The map was deposited in the electron microscopy data bank (EMDB) with ID EMD-42981, and the atomic model in the protein data bank (PDB) with ID 8V5A. The sequences of RV3 F wildtype, OnlyEcto2P and OnlyEcto have been made available in sFig 9. Source data are provided with this paper.

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

## Acknowledgements

We thank Anne Thoma, Lisanne Tettero, Cornelis Vaneman, Jan Serroyen and Tina Ritschel for technical assistance and fruitful discussions.

## Author contributions

J.P.M.L., F.C. and M.J.G.B. designed the study. J.J. and M.J.G.B. designed F variants. F.C., D.v.O., L.L., W.v.d.H., R.V., L.v.d.F, R.Z. and M.J.G.B. planned and / or performed immunological and biochemical assays and purifications or in vivo studies. N.V.J. and J.S.M. performed the Cryo-EM characterization. J.P.M.L., F.C., N.V.J. and M.J.G.B. wrote the paper with input from all other authors.

## Competing interests

J.P.M.L., J.J., and M.J.G.B. are co-inventors on related vaccine patents. J.P.M.L., F.C., D.v.O., L.L., W.v.d.H., R.V., R.Z., L.v.d.F., J.J. and M.J.G.B. are employees of Janssen Vaccines & Prevention BV and may hold stock of Johnson and Johnson.
