## [Peer Review File · Nature Communications]

nature portfolio

Peer Review File

Editorial Note: Parts of this peer review file have been redacted as indicated to maintain the confidentiality unpublished data.REVIEWER COMMENTS

Reviewer #1 (Remarks to the Author):

Langedijk et al. report a new way to stabilize the F protein of respirovirus 3 (RV3), an important human pathogen for which neither vaccine nor therapeutics exist. Furthermore, they show that the approach is applicable to stabilization of the F protein from a more distant paramyxovirus such as Nipah virus, which has high mortality rate in humans. A systematic approach to introducing mutations into F protein has been applied, allowing the authors to identify combinations of mutations that have synergistic effect. 54 F variants have been evaluated comprehensively for the expression levels, stability, and binding to antibodies specific for the pre-fusion conformation of the protein. The structure of the best candidate, OnlyEcto, has been determined by cryo-EM providing structural explanations for the effect of the engineered mutations on stability and activity of the protein. The results are noteworthy in particular because a set of stabilizing mutations targets the HRB region, which has not been investigated much before. The authors show that the HRB mutations in combination with others stabilize the F protein sufficiently so that the heterologous and so-far necessary trimerization motif (GCN4 or foldon) is not required anymore. This result has important implications for vaccine design, where heterologous motifs provide unwanted off-target epitopes. The best candidate, OnlyEcto variant, was then shown to induce strong immunogenicity and elicit neutralizing antibodies in mice, naïve or pre-exposed, and to provide protection in cotton rat challenge model. These findings are important and provide a significant advance in the fields of immunogen design and vaccine development against paramyxoviruses in general. It was a pleasure reading this manuscript.

Major remarks

1.The supplementary information file has not been included if I am not mistaken.

2.The introduction would benefit from a clarification on what has been tried on paramyxovirus F in terms of stabilization before. This would highlight the importance of the experiments in this manuscript i.e. targeting HRB and systematically testing proline substitutions, which to the knowledge of this reviewer has not been done before.

3.A table summarizing all the tested mutants would help the reader keep track of all variants tested, and their expression levels / stability. The table would ideally have: the name of the construct (OnlyEcto, OnlyEcto2P, Ecto4P etc), the list of all mutations, how well expressed the protein was, how stable it was, and what kind of neutralizing response it elicited.

4.Along the previous remark, sequences of all the constructs, in the form of protein sequences listed in a Table, or plasmid files, should be provided as a supplement with the manuscript. While this

reviewer is aware that this requires some tedious work, it is necessary. It will help the readers, and anyone interested in expressing the proteins and reproducing results.

5. Validation reports for the cryo-EM structures are not in the right format (they are clearly labelled as 'not for the manuscript review') and are missing some important information in section 4 "Experimental information". It is not clear how the authors achieved 3.1 Å resolution mentioned on line 203 (the validation report states differently). There is probably more information in sTable 1, which could not be found.

Minor remarks

1. Figure and figure legends need some improvement:

a. Figure 1B: could the authors explain why the V_0 (initial slope) was used as an indication of protein amount instead of for example the BLI signal achieved when the binding is saturated. Measuring the rate implies how fast the protein binds, and (intuitively) does not seem to necessarily reflect the amount of protein present in the supernatant. But assuming there a reason exists for using V_0 , units should be included, presumably 1/s, and the V_0 should be clearly defined.

b. Figure 3: considering that the manuscript will be of interest to structural biologist, some not used to look at the fusion assay images, an illustration of what we are trying to grasp would be helpful. Maybe explain what a syncytium is and how it is spotted, in the positive control for example. And what is blue and what is pink?

c. Figure 4A, B: adding a color curve legend of the panel would help (purple – 4P, grey – WT, blue - ... etc)

d. Figure 4C, D: could the authors think of a different way to label the y-axis so it reflects what is measure better? State what the '1st derivative' is of and why use the 1st derivative instead of raw signal, and for the 'shift' (nm) on panel D, specify what shift is measure and what it reflects (in figure legend, if not on the graph).

e. Figure 4H: it is hard to see that the 4 curves are overlaid on the left panel. Maybe a zoom at the top of the curves could be added as an inset?

f. Figure 5: panels C and E are swapped. On current panel C, Q222I should be Q221I.

g. Figure 6 – panels C, D – units on y-axis should be added.

h. Figure 7 – could the authors define in the legend or text what is plotted on y-axes. For example, what is 'Rel Pot titer', 'IC50 titer'? Define abbreviations VNA and PRNT, and make it more friendly to non-immunologists perhaps.

i. Figure 8: panel A – to be consistent, maybe stick to HRA and HRB instead of HR1 and HR2. Domains I and II are not colored in yellow in the postfusion panel. It is not clear what the prefusion structure is of – is it the F variant whose structure was determined here (OnlyEcto?) or it is another, already reported structure; panel B – the yellow sphere for S41 is very hard to see. Panel E – structural elements shown here should be labelled. It is not clear what we are looking at.

2. Have the authors generated and checked the AF2 predicted model for the OnlyEcto trimer. It could be interesting to see if the mutated residues would be predicted correctly by AF2.

3. Line 120 – to be precise, could it be stated in terms of residue x-y what the authors define as HRB and ‘HRB stem’?

4. Line 317 – “... significantly higher...” – could the authors use a number instead the qualitative description?

5. Line 321 – EctoOnly should be OnlyEcto

6. Line 366 – ‘the long β sheet connecting DI and DIII’. This sheet is a part of domain III. Could the authors rephrase the sentence?

7. Sentence, lines 545-546, is not clear.

8. Could the authors provide more information (manufacturer) on adjuvant AS01B (line 593), and Adjuphos (line 294)?

Reviewer #2 (Remarks to the Author):

Well-written and straight forward paper on stabilization of paramyxovirus F protein in the prefusion conformation, without the need for a trimerization domain. The approach was shown to be applicable for multiple paramyxovirus F proteins.

Remarks:

1. Figure 1. cell culture supernatants are screened using octet and a prefusion-specific antibody. Differences in antibody binding may result from differences in 1) F protein stability (and thus conformation), 2) differences in expression levels, 3) substitutions in the epitope. Probably the authors can rule out option 3 based on the literature, but not necessarily option 2. This should be clarified.

2. Line 163. FACS analysis indicate similar expression levels, but if the F proteins differ in stability/conformation, may it not be that staining with the prefusion-specific antibody is not corresponding with expression levels per se?
3. Figure 6. Are the antibodies used conformation dependent/prefusion specific? How can the authors discriminate between differences in expression levels or conformation for the different F proteins.
4. Line 292: how did the authors check that postF is post F.
5. Can the authors comment on the expression levels and suitability for large scale protein of vaccine antigens, might additional modifications needed?

Reviewer #3 (Remarks to the Author):

In this manuscript. Langedijk et al. designed an experimental subunit vaccine for respirovirus 3 (RV3, previously HPIV3) based on a stabilized fusion protein (preF) as previously done for HRSV and SARS-CoV-2. They evaluated multiple stabilizing mutations in different experimental set-ups, and combined these to generate a vaccine candidate. Vaccination of mice and cotton rats with a preF-stabilized protein (OnlyEcto) led to induction of (neutralizing) antibodies and protected cotton rats from lower respiratory tract infection if adjuvants were used. Additionally, the authors show that they were able to transfer the stabilizing mutations to RV1-F and NiV-F proteins. This is a complete study (antigen design to in vivo), which is based on strong data and clearly written. I have several suggestions towards improving clarity.

Major comments

1. Throughout the manuscript, the study of certain (combinations of) mutations should be better rationalized to help the reader follow the authors' decisions.
 - a. Lines 126-133: Why did the authors chose for Q89M+Q222I and L168P to test the 470/477 mutation? Was this because the BLI binding rate was the highest?
 - b. Line 134: The authors introduced both the 470 and 477 mutation (on top of the five head-stabilizing mutations) based on temperature stability; Figure 2C and line 132 suggest that only S470V leads to better SEC data. The explanatory temperature data shown in Figure 2E should be moved forward.

c. Figure 3 and lines 155–165: Based on the story, I would expect to see fusion data on the F protein with the 7 introduced mutations described in Figure 2 (S41P, N167P, Q89M+Q222I, L168P, S470V, S477V). Additionally, OnlyEcto2P and OnlyEcto should have been evaluated. Are those available?

d. Line 169: Why was F335P added to the previously described 7 mutations to generate OnlyEcto2P? This is relevant, as F335P is later reversed.

2. At the end of the paragraph ‘Stabilized RV3 F is in the prefusion confirmation and retains neutralizing epitopes’ corresponding to Figure 4, the reader gets the impression that OnlyEcto is the preferred candidate. Why is all cryoEM performed with OnlyEcto2P instead? At the same time the authors are describing the function of F335 and N167, although these are not part of the vaccine candidate.

3. Line 248 and Figure 6D: For RV1 F, why was I168P not introduced into the multi-mutation F protein considering the binding rate being higher than E170P which was introduced.

4. Lines 270-273 and Figure 6I,J: Why is K167P not referred to as the mutation with the highest expression according to BLI and SEC and therefore investigated further? The SEC line is difficult to see in panel J but looks similar to A165P.

5. Line 414-416: the authors suggest that their preF candidate is able to protect from severe disease associated with LTI. They should be careful with their conclusions: the performed experiment does not reflect disease or pathology, and was performed in an optimal manner (two vaccinations, followed by a challenge after 3 weeks). This could be put in better perspective.

6. Although the mutations in general are transferable to NiV, the authors do face some expression issues. These expression issues are hardly discussed, and this should be added to the manuscript. At the same, is it warranted to use the word universal in the title?

7. The authors should consider exchanging Figure 6 and Figure 7 and the accompanying paragraphs to show functional data for RV3, before transferring the system to other paramyxoviruses.

8. The authors suggest a role for neutralizing antibodies as correlate of protection in Figure 7; however, no other immunological parameters were measured, making this an overstatement. I suggest removing the data on correlates of protection. Lines 424-431: The authors mention T-cells in the paragraph before; however, it should be mentioned next to the many advantages of protein vaccines that they are not good cytotoxic T-cell inducers.

Minor comments

1. In the abstract the authors state that no RV3 vaccine is available, and next state the preF stabilization is necessary for vaccine effectiveness. This is confusing.

2. Line 75: the authors suggest that the absence of a heterologous trimerization domain is advantageous, but only later explain why. Would make more sense to explain that here.

3. Line 183: (Johnson et al. in preparation) suggests that the data are not published. This should be rephrased.
4. Line 207: remove parenthesis after 473-484.
5. Line 216: the authors have continuously used the names of the mutations. Therefore, 'double proline' should be renamed.
6. Line 244 and 265/266: the authors mention that stabilizing substitutions for the RV1-F and NiV-F protein are based on the RV3 protein. It would benefit the reader if the mutations shown in Figure 6D and 6I could be matched to the RV3-F substitutions. From the manuscript, this is only possible for S41P.
7. Lines 267/268: Panel J is mentioned before panel I.
8. Lines 294/295: 'higher levels' of antibodies compared to mock suggests that mock animals had antibodies, which is not the case. This should be re-phrased.
9. Line 321: EctoOnly should be OnlyEcto.
10. Lines 324-327 and methods section: The pre-exposure of cotton rats with RV3 is not described.
11. Lines 413 and 414: 'protection' should be complemented with 'from infection'. The authors did not show pathology data for the upper or lower respiratory tract.
12. Lines 587-598: How were pre-challenge blood samples from cotton rats obtained?
13. In Figure 1C, the authors should define a cut-off to define their most favorable combinations of mutations. This would help the reader to follow why for example S41P + F335 was not initially selected despite a beneficial effect of the combination.
14. Figure 4 would benefit from figure legends. In which panel does the reader look at OnlyEcto2P or OnlyEcto data?
15. Figure 4J: Should the mutations not be reversed? Δ S41P suggests that the difference from S to P is presented here but the panel shows the difference after reversion from P to S.
16. Figure 5A is not referred to in the manuscript.
17. Figure 6 would benefit from figure legends. In which panel does the reader look at RV1 and where at Nipah virus? For the SEC data, the corresponding F substitutions could be mentioned.
18. sFig7A-C data could be added to Figure 6 of the main manuscript.
19. Figure 7A,B, E, G: the abbreviation Rel Pot titer should be explained in the caption.
20. Figure 7C, F, H and methods section: How was the IC50 titer determined? Was this a dose-response curve and would VN titer not be the more appropriate axis label?
21. Figure 7I and J would benefit from the figure legend from panel K.

22. In both the txt and figures, sometimes the authors refer to viruses (for example RV3), sometimes to diseases (measles and nipah, instead of MeV and NiV). The authors should be consistent.

REVIEWER COMMENTS

Reviewer #1 (Remarks to the Author):

Langedijk et al. report a new way to stabilize the F protein of respirovirus 3 (RV3), an important human pathogen for which neither vaccine nor therapeutics exist. Furthermore, they show that the approach is applicable to stabilization of the F protein from a more distant paramyxovirus such as Nipah virus, which has high mortality rate in humans. A systematic approach to introducing mutations into F protein has been applied, allowing the authors to identify combinations of mutations that have synergistic effect. 54 F variants have been evaluated comprehensively for the expression levels, stability, and binding to antibodies specific for the pre-fusion conformation of the protein. The structure of the best candidate, OnlyEcto, has been determined by cryo-EM providing structural explanations for the effect of the engineered mutations on stability and activity of the protein. The results are noteworthy in particular because a set of stabilizing mutations targets the HRB region, which has not been investigated much before. The authors show that the HRB mutations in combination with others stabilize the F protein sufficiently so that the heterologous and so-far necessary trimerization motif (GCN4 or foldon) is not required anymore. This result has important implications for vaccine design, where heterologous motifs provide unwanted off-target epitopes. The best candidate, OnlyEcto variant, was then shown to induce strong immunogenicity and elicit neutralizing antibodies in mice, naïve or pre-exposed, and to provide protection in cotton rat challenge model. These findings are important and provide a significant advance in the fields of immunogen design and vaccine development against paramyxoviruses in general. It was a pleasure reading this manuscript.

We would like to thank the reviewer for taking the time to review our manuscript and greatly appreciate the positive feedback and insightful comments. We are delighted to hear that you appreciated the significance of our work on stabilizing the F protein of RV3 and its application to Nipah virus. Please find detailed answers to your questions below.

Major remarks

1. The supplementary information file has not been included if I am not mistaken.

That is unfortunate, the supplementary figures and legends were supposed to be attached as PDF document. We have reuploaded them to be certain.

2. The introduction would benefit from a clarification on what has been tried on paramyxovirus F in terms of stabilization before. This would highlight the importance of the experiments in this manuscript i.e. targeting HRB and systematically testing proline substitutions, which to the knowledge of this reviewer has not been done before.

The reviewer makes an excellent point; we have now included a description of previous stabilization efforts in the introduction (lines 70-73).

3. A table summarizing all the tested mutants would help the reader keep track of all variants tested, and their expression levels / stability. The table would ideally have: the name of the construct (OnlyEcto, OnlyEcto2P, Ecto4P etc), the list of all mutations, how well expressed the protein was, how stable it was, and what kind of neutralizing response it elicited.

We appreciate the suggestion by the reviewer and improved the clarity of the OnlyEcto2P versus OnlyEcto comparison by additional labels in figure 4 to depict more clearly which data belongs to OnlyEcto2P and which to OnlyEcto. As we only have detailed biochemical analysis of two 'lead' designs, the only relevant comparison can be made for these two variants. The outcome of all these characterizations is shown side-by-side in Fig 4. The screening data described in the manuscript is based

on single or double mutations and their data is summarized in a comprehensible format in Fig. 1B, C.

4. Along the previous remark, sequences of all the constructs, in the form of protein sequences listed in a Table, or plasmid files, should be provided as a supplement with the manuscript. While this reviewer is aware that this requires some tedious work, it is necessary. It will help the readers, and anyone interested in expressing the proteins and reproducing results.

We agree wholeheartedly that our publication and similar publications in the public domain would benefit from easy access to the sequences. To this end we have added the sequences of RV3 F wildtype, OnlyEcto2P and OnlyEcto in novel supplementary Fig 9. We have highlighted the stabilizing substitutions and C-tag design in order to make it straightforward to obtain the sequences of our leads, and any partially stabilized derivatives thereof.

5. Validation reports for the cryo-EM structures are not in the right format (they are clearly labelled as 'not for the manuscript review') and are missing some important information in section 4 "Experimental information". It is not clear how the authors achieved 3.1 Å resolution mentioned on line 203 (the validation report states differently). There is probably more information in sTable 1, which could not be found.

We apologize for erroneously uploading the wrong validation report. We have now included the final report and re-uploaded the supplementary data, including supplementary table 1. Regarding the resolution that we state in the manuscript we would like to note that we report the resolution that is calculated based on a tight mask with correction by noise substitution. On the other hand, the resolution calculated by the validation report is the unmasked resolution. Please note that we report both the masked and unmasked resolutions in the supplemental table.

Minor remarks

1. Figure and figure legends need some improvement:

a. Figure 1B: could the authors explain why the V_0 (initial slope) was used as an indication of protein amount instead of for example the BLI signal achieved when the binding is saturated. Measuring the rate implies how fast the protein binds, and (intuitively) does not seem to necessarily reflect the amount of protein present in the supernatant. But assuming there a reason exists for using V_0 , units should be included, presumably 1/s, and the V_0 should be clearly defined.

The reviewer raises a valid point and actually hits upon a general difficulty that we often face in the early stages of the design process that has to do with low expression of unmodified and minimally stabilized variants. Ideally, we use analytical Size-exclusion chromatography to get a definitive answer on protein expression in clarified supernatant. Unfortunately, this technique requires a higher base level of protein expression for detection above the mock signal compared to BLI. Most of the variants in Fig 1B do not reach sufficient levels for detection by analytical SEC. So instead, we resorted to BLI and the use of the binding rate as measured over the full association phase. Since we saturate the sensor tip with PIA174, an antibody that binds tightly to the RV3 F apex, the end point shifts tend to reach similar values for all variants, irrespective of expression levels. On the other hand, the initial binding rate increases proportionally to the amount of preF protein in the supernatant and is thus a more accurate measurement of expression levels. We have now added more details on the read-out in the BLI paragraph of the Materials & Methods section, additionally we have added the units on the y-axis (nm/s) throughout the figures.

b. Figure 3: considering that the manuscript will be of interest to structural biologist, some not used to look at the fusion assay images, an illustration of what we are trying to grasp would be helpful. Maybe explain what a syncytium is and how it is spotted, in the positive control for example. And what is blue and what is pink?

We realize that the cell-cell fusion assay might be unknown to a portion of the readers. We have now included a more detailed description of the read-out (including the colors) in the method section (lines 530-534) and legend of Fig 3, and included a brief definition of a syncytium in the results section (lines 154-155).

c. Figure 4A, B: adding a color curve legend of the panel would help (purple – 4P, grey – WT, blue - ... etc) We like to thank the reviewer for this suggestion. We have now added a legend in the figure and repaired the '4P' typo in the legend, this should have been 'OnlyEcto2P'.

d. Figure 4C, D: could the authors think of a different way to label the y-axis so it reflects what is measured better? State what the '1st derivative' is of and why use the 1st derivative instead of raw signal, and for the 'shift' (nm) on panel D, specify what shift is measured and what it reflects (in figure legend, if not on the graph).

We have improved the labelling of the y-axis throughout the figures and added additional details to the methods section (lines 514-515). The melting temperatures are derived from the negative first derivative of the fluorescent signal which we plotted as a function of temperature. The lowest point in the curve indicates the T_{m50} . We prefer this way of plotting the data since we believe it is more intuitive to observe the tipping point for the 1st derivative (example below, right panel) than it is for the raw fluorescent signal (example below, left panel).

For the BLI measurements in Fig 4D, E, we choose to plot the raw binding curves as measured by the Octet instrument. This will allow readers to see that any differences in mAb binding are not due to differences in amount of preF captured on the sensor. We have added additional details to the figure legend (lines 866-868)

e. Figure 4H: it is hard to see that the 4 curves are overlaid on the left panel. Maybe a zoom at the top of the curves could be added as an inset?

We thank the reviewer for this suggestion and have implemented the change in Fig 4H.

f. Figure 5: panels C and E are swapped. On current panel C, Q222I should be Q221I.

We thank the reviewer for this keen observation. We have swapped the legends so that they are correct once again. There was indeed also a typo in the legend: the double substitution should have been Q89M+Q222I, some confusion might have arisen because we also found a stabilizing disulfide in the same region (G85C-Q221C).

g. Figure 6 – panels C, D – units on y-axis should be added.

Indeed, we have now added the units (nm/s) to the BLI data throughout the figure set.

h. Figure 7 – could the authors define in the legend or text what is plotted on y-axes. For example, what is 'Rel Pot titer', 'IC50 titer'? Define abbreviations VNA and PRNT, and make it more friendly to non-immunologists perhaps.

To improve readability, we have made changes to Fig 7, specifically by improving the y-axis legends. For instance, we replaced "Rel Pot titer" with "Binding IgG titer" and "Neutralizing antibody titer".

Additionally, we have included and defined abbreviations in the figure legends where relevant.

i. Figure 8: panel A – to be consistent, maybe stick to HRA and HRB instead of HR1 and HR2. Domains I

and II are not colored in yellow in the postfusion panel. It is not clear what the prefusion structure is of – is it the F variant whose structure was determined here (OnlyEcto?) or it is another, already reported structure; panel B – the yellow sphere for S41 is very hard to see. Panel E – structural elements shown here should be labelled. It is not clear what we are looking at.

We modified the figure to now read HRA and HRB to improve consistency. The structure is indeed that of OnlyEcto2P that we here describe; we have now added this information to the legend. The postfusion structure is based on PDB ID 1ZTM. This is indicated in the legend and a reference is added. The yellow spheres are indeed somewhat hard to see. Since they make up the corners of the red triangle we added ‘at the corner of the triangle’ in the legend to make it more clear to readers. We have now clarified panel E by adding labels for the two structural elements (DI-DIII β -sheet and the helical bundle (HB)). Moreover, we added a figure reference to the structural elements relative to the rest of the Pre-F trimer in line 399.

2. Have the authors generated and checked the AF2 predicted model for the OnlyEcto trimer. It could be interesting to see if the mutated residues would be predicted correctly by AF2.

We had not run AF2 prior to this question. When we generated an AF2 model for OnlyEcto, we saw that it generally aligned well to either the helical domains composed of HRC, HRA, and domain II OR to the region dominated by beta sheets composed of domains I and II. However, it placed these regions poorly relative to each other. Unlike many other class I fusion proteins, the paramyxovirus F trimer forms a large central cavity that is not well accounted for in the AF2 model, leading to an apparent collapse of the trimer into a more compact conformation. When either region of the AF2 model was aligned to the corresponding region in the cryo-EM structure, the positions of single substitutions were generally well predicted. Specifically, the position of S41P within B1 was predicted to be slightly rotated relative to the cryo-EM structure, as this position marks the border between domains III and I within the long B1 helix. Similarly, the predicted position of F335P within the loop preceding B12 was only slightly shifted relative to the cryo-EM structure when domains I and II were aligned. The HRA substitutions N167P and L168P were also reasonably well predicted by the AF2 model, both within the B3/B4 loop and in relative position to proximal residues within B1. Interestingly, while the Q222I substitution revealed a short, ordered helix that had been previously unresolved (and is likely flexible and disordered in the wild-type protein), this helix was not predicted in any of the AF2 models. As a result, Q222I was shifted away from Q89M and the Ile sidechain is pointed away from the HRC helix. This also resulted in a positional shift of the α 6, α 7, and α 8 helices, which aligned more poorly than the rest of domain III. Lastly, S470V and S477V were predicted to be in the same position within the HRB helix in the AF2 model as determined by cryo-EM.

3. Line 120 – to be precise, could it be stated in terms of residue x-y what the authors define as HRB and ‘HRB stem’?

We added the exact residues and reworded the sentence to improve clarity: “Next, the inherently labile RV3 HRB domain (residues 447-484) that forms the stem (residues 453-484), was stabilized...”

4. Line 317 – “... significantly higher...” – could the authors use a number instead the qualitative description?

We would like to thank the reviewer for this excellent suggestion. We have now added the fold increase of the preF binding and RV3 neutralizing antibody titers of OnlyEcto immunized mice relative to the pre-exposed non-immunized mice (lines 324-325).

5. Line 321 – EctoOnly should be OnlyEcto

Thank you for bringing this mistake to our attention; we have made the necessary correction.

6. Line 366 – ‘the long β sheet connecting DI and DIII’. This sheet is a part of domain III. Could the authors rephrase the sentence?

The reviewer is correct, we have now reworded the sentence: "...the long β -sheet of DIII that connects to DI, and an inner layer that..."

7.Sentence, lines 545-546, is not clear.

This sentence was indeed incorrect; we have now rephrased it: "Purified RV3 preF (OnlyEcto2P) at 1 mg/mL was combined with 1.25 molar excess VHH 4C06 in buffer containing 2 mM TRIS pH 8.0, 200 mM NaCl, 0.02 % NaN₃, and 0.01 % amphipol."

8.Could the authors provide more information (manufacturer) on adjuvant AS01B (line 593), and Adjuphos (line 294)?

AS01B, the adjuvant component of Shingrix manufactured by GSK (GlaxoSmithKline, London, UK), was separately provided from the vaccine and was used in the cotton rat study. As for the Alum adjuvants, we used AIPO which was sourced from Invivogen (Adju-Phos) or AIOH which was manufactured in-house by Janssen. The main text now includes clear mention of the specific source of Alum adjuvant used. This information, including the manufacturers, is given in the Methods section of the manuscript, in the sections describing the animal experiments.

Reviewer #2 (Remarks to the Author):

Well-written and straight forward paper on stabilization of paramyxovirus F protein in the prefusion conformation, without the need for a trimerization domain. The approach was shown to be applicable for multiple paramyxovirus F proteins.

We thank the reviewer for these kind words; please find the answers to your questions below.

Remarks:

1. Figure 1. cell culture supernatants are screened using octet and a prefusion-specific antibody. Differences in antibody binding may result from differences in 1) F protein stability (and thus conformation), 2) differences in expression levels, 3) substitutions in the epitope. Probably the authors can rule out option 3 based on the literature, but not necessarily option 2. This should be clarified. Indeed, the reviewer raises a valid point. No stabilizing substitutions in, or close to, the PIA174 epitope were screened, so option 3 can be ruled out. Regarding options 1 and 2; it is our experience that protein stability and expression levels go hand-in-hand. For vaccine manufacturing, both qualities are important and therefore, substitutions are selected that improve expression and improve stability. In the following evaluation rounds using purified protein, the impact on stability of the combination of mutations are more accurately tested.

2. Line 163. FACS analysis indicate similar expression levels, but if the F proteins differ in stability/conformation, may it not be that staining with the prefusion-specific antibody is not corresponding with expression levels per se?

The reviewer raises an excellent point. However, if anything, the stabilizing mutations would likely lead to higher expression. Nonetheless, on the confined dimensions of a cell membrane we usually observe a less profound impact of prefusion stabilization on expression. Despite the similar or higher expression levels, still no fusion is observed. To strengthen the manuscript, we have now included an additional FACS staining using a panF-binding antibody, which also shows no major differences between constructs in expression levels (sFig 1, right panel).

3. Figure 6. Are the antibodies used conformation dependent/prefusion specific? How can the authors discriminate between differences in expression levels or conformation for the different F proteins. The antibodies used in BLI, 3X1 (for RV-1 F) and 5B3 (for NiV F) are indeed conformation-, and preF-specific, we have indicated this in lines 247 and 267, of the updated manuscript, respectively. The aim is

to improve preF expression levels, so initially we focus on an increase in preF-specific antibody binding, and later on, when expression levels are sufficient, we continue using analytical SEC because this is a more quantitative method.

4. Line 292: how did the authors check that postF is post F.

The protein was designed and purified as described previously for RV-3 postF (Stewart-Jones et al PNAS 2018). The purified protein gave a single peak on SEC-MALS (see Fig below, *left*) with a shorter retention time than RV-3 preF (compare e.g. to Fig 4F), due to the elongated shape of the postF conformation. Moreover, we did not observe binding to preF-specific mAb PIA174, while we did observe binding to postF-specific Ab mAb200105) and panF-binding Ab mAb200104 in BLI (see Fig below, *right*). Please note that the postF specific Ab will be part of a future publication and we therefore choose not to include this data in the manuscript.

[redacted]

5. Can the authors comment on the expression levels and suitability for large scale protein of vaccine antigens, might additional modifications needed?

Expression levels are exceptionally high (>330mg/L) in transient Expi293 cell systems, this is 5-10 fold higher than we observe for other stabilized paramyxovirus and pneumovirus F proteins. These levels are well beyond the lower expression limits required for a viable commercial vaccine, so it is not expected that additional modifications will be necessary.

Reviewer #3 (Remarks to the Author):

In this manuscript. Langedijk et al. designed an experimental subunit vaccine for respirovirus 3 (RV3, previously HPIV3) based on a stabilized fusion protein (preF) as previously done for HRSV and SARS-CoV-2. They evaluated multiple stabilizing mutations in different experimental set-ups, and combined these to generate a vaccine candidate. Vaccination of mice and cotton rats with a preF-stabilized protein (OnlyEcto) led to induction of (neutralizing) antibodies and protected cotton rats from lower respiratory tract infection if adjuvants were used. Additionally, the authors show that they were able to transfer the stabilizing mutations to RV1-F and NiV-F proteins. This is a complete study (antigen design to in vivo), which is based on strong data and clearly written. I have several suggestions towards improving clarity.

We would like to thank the reviewer for the invested time in reviewing our manuscript and the constructive feedback. We sincerely appreciate your positive feedback on our work regarding the design of an experimental subunit vaccine for RV3. Your suggestions for improving clarity are invaluable, and we have incorporated them to enhance the quality of our manuscript. Please find the answers to your questions and suggestions below.

Major comments

1. Throughout the manuscript, the study of certain (combinations of) mutations should be better rationalized to help the reader follow the authors' decisions.

a. Lines 126-133: Why did the authors chose for Q89M+Q222I and L168P to test the 470/477 mutation? Was this because the BLI binding rate was the highest?

Indeed, we picked this combination because it gave the highest expression in presence of a GCN4 trimerization domain (Fig 1C). We have now included this rationale in the revised manuscript: "An RV3 F variant with three stabilizing substitutions in the head domain (Q89M+Q222I and L168P) gave the highest trimer expression in presence of GCN4 (Fig. 1C), but only showed low expression of F monomers when the GCN4 trimerization domain was removed, as determined by analytical size-exclusion chromatography (SEC) on cell culture supernatant of transfected Expi293F cells (Fig. 2C, blue line)." (lines 130-135).

b. Line 134: The authors introduced both the 470 and 477 mutation (on top of the five head-stabilizing mutations) based on temperature stability; Figure 2C and line 132 suggest that only S470V leads to better SEC data. The explanatory temperature data shown in Figure 2E should be moved forward. The backbone used in Fig 2C only expresses as monomers, and we find that individual introduction of 470V or 477V is sufficient to produce trimers. So, while S470V has the greatest impact on trimer expression, S477V was also selected based on the SEC data. We realize the original text was somewhat confusing and have reworded the sentence to better reflect the data: "When the 470 and 477 positions were optimized by substitution of the individual or combined Ser residues for Val residues to allow more favorable interactions (Fig. 2A, B), it resulted in a sharp increase in trimer expression in each case (Fig. 2C)."

c. Figure 3 and lines 155–165: Based on the story, I would expect to see fusion data on the F protein with the 7 introduced mutations described in Figure 2 (S41P, N167P, Q89M+Q222I, L168P, S470V, S477V). Additionally, OnlyEcto2P and OnlyEcto should have been evaluated. Are those available?

We like to thank the reviewer for this suggestion. However, we do not believe that the combinations will be more informative than the variants containing single mutations, given that S41P, L168P and Q89M+Q222I individually already fully block fusion, and none of the other substitutions enhance fusogenicity. Therefore, we can conclude that any further combinations, like EctoOnly and EctoOnly2P, that contain these fusion-preventing substitutions mentioned above will likewise be fusion inactive.

d. Line 169: Why was F335P added to the previously described 7 mutations to generate OnlyEcto2P? This is relevant, as F335P is later reversed.

F335P was added because it was the best stabilizing proline substitution in our initial screen (Fig 1B), was compatible with the other 7 substitutions we had selected (Fig 1C) and by itself could fully block cell-cell fusion (Fig. 3). We have added this rationale to the results section (lines 175-177). The reasoning for the subsequent reversal of F335P is detailed in lines 189-193 of the revised manuscript.

2. At the end of the paragraph 'Stabilized RV3 F is in the prefusion confirmation and retains neutralizing epitopes' corresponding to Figure 4, the reader gets the impression that OnlyEcto is the preferred candidate. Why is all cryoEM performed with OnlyEcto2P instead? At the same time the authors are describing the function of F335 and N167, although these are not part of the vaccine candidate.

Based on antigenicity analysis by BLI and SEC-MALS analysis we can conclude that both designs are in the desired preF conformation. We decided to solve the structure of OnlyEcto2P because it has additional mutations and a structure of those would be informative. This is especially the case for F335P, since it has such a profound impact on Tm50. However, for a vaccine we need to consider and balance expression and stability on the one hand, and antigenicity on the other. In the end we decided to move forward with OnlyEcto as to have a minimal impact on the immunogen as described above.

3. Line 248 and Figure 6D: For RV1 F, why was I168P not introduced into the multi-mutation F protein considering the binding rate being higher than E170P which was introduced.

We choose to include E170P and Q171P because they are the equivalents to the N167P and L168P substitutions we applied in EctoOnly2P. The reviewer is correct to assume that other combinations of stabilizing substitutions are also possible, however, these were not tested at this stage.

4. Lines 270-273 and Figure 6I,J: Why is K167P not referred to as the mutation with the highest expression according to BLI and SEC and therefore investigated further? The SEC line is difficult to see in panel J but looks similar to A165P.

Indeed, K167P has a similar or perhaps slightly higher effect on expression levels as A165P. We selected the latter because it is a much more conservative change and less likely to destroy potential antibody epitopes. We have now rephrased the sentence and added a short rationale: "Expression of the variant with the conservative A165P substitution in combination with S470V and A477V gave among the highest expression levels according to both BLI and analytical SEC..." (lines 279-281).

5. Line 414-416: the authors suggest that their preF candidate is able to protect from severe disease associated with LTI. They should be careful with their conclusions: the performed experiment does not reflect disease or pathology, and was performed in an optimal manner (two vaccinations, followed by a challenge after 3 weeks). This could be put in better perspective.

We acknowledge this critical point raised by the reviewer. Despite our efforts in choosing our wording thoughtfully, we understand that it may have unintentionally conveyed a different implication.

Consequently, we have removed the part of the phrase "severe disease associated with" from that specific line to avoid any potential over-interpretation of our data.

6. Although the mutations in general are transferable to NiV, the authors do face some expression issues. These expression issues are hardly discussed, and this should be added to the manuscript. At the same, is it warranted to use the word universal in the title?

The reviewer is correct that the base expression levels of Nipah preF are lower than that of PIV3 preF. Based on these successful transfers of our RV3 F stabilizing substitutions to RV1 and Nipah, we believe that the universality claim is warranted. Please note that we hoped to address the lower NiV F expression levels in panel 6L, where we show that they can be further boosted by additional substitutions in HRB, with S466D, Q469E, Y473E and L480K boosting expression of NiV preF levels to PIV3 preF levels.

7. The authors should consider exchanging Figure 6 and Figure 7 and the accompanying paragraphs to show functional data for RV3, before transferring the system to other paramyxoviruses.

We appreciate the suggestion and did indeed consider swapping the two paragraphs during preparation of the manuscript. In the end we decided on the current order since it would conclude all the design and structural work, before moving on to the preclinical evaluation.

8. The authors suggest a role for neutralizing antibodies as correlate of protection in Figure 7; however, no other immunological parameters were measured, making this an overstatement. I suggest removing the data on correlates of protection. Lines 424-431: The authors mention T-cells in the paragraph before; however, it should be mentioned next to the many advantages of protein vaccines that they are not good cytotoxic T-cell inducers.

We would like to thank the reviewer for this careful suggestion. In the discussion of the manuscript, we suggest a potential role for neutralizing antibody responses in protection based on the robust correlation observed between neutralization and protection. We believe that showcasing these data is important as it might provide a clue about the underlying mechanism responsible for the observed protection. We do agree with the reviewer that "emphasizing" is too strong a statement, we have now replaced it with "suggesting" (line 429 of the revised manuscript). It should be noted that we did not measure cellular immune responses in mice as unadjuvanted protein vaccines are not typically known for their capability to induce such responses. However, in the cotton rat study, AS01b adjuvant was used

that has been demonstrated to enhance cellular immune responses. To provide better context to our suggestion that neutralizing antibody responses might play an important role in the observed protection, we have included the following statement in the discussion: "To further understand the immunological mechanism of protection mediated by RV3 preF based vaccines, it is necessary to conduct additional characterization of the antibody and cellular immune responses, and explore their interplay." (lines 432-434).

Minor comments

1. In the abstract the authors state that no RV3 vaccine is available, and next state the preF stabilization is necessary for vaccine effectiveness. This is confusing.

Indeed, we state in the abstract that no RV3 vaccine has been approved for use in humans, however a preclinical experiment with a preF subunit vaccine candidate by Stewart-Jones et al (PNAS 2018) has been published previously. This experiment showed that preF induced higher neutralizing antibody responses compared to postF. This concept has also been shown extensively for the pneumoviruses RSV and HMPV and is accepted in the field. We have reworded the sentence in the abstract to be more clear: "The RV3 fusion (F) protein is inherently metastable and will likely require prefusion (preF) stabilization for vaccine effectiveness."

2. Line 75: the authors suggest that the absence of a heterologous trimerization domain is advantageous, but only later explain why. Would make more sense to explain that here.

The reviewer raises a valid point; we now added the rationale here as well: "and will therefore not induce off-target responses".

3. Line 183: (Johnson et al. in preparation) suggests that the data are not published. This should be rephrased.

This publication will likely be published before this manuscript and the reference will be updated accordingly.

4. Line 207: remove parenthesis after 473-484.

Done

5. Line 216: the authors have continuously used the names of the mutations. Therefore, 'double proline' should be renamed.

Agreed, we have reworded to state the identity of the mutations throughout the manuscript.

6. Line 244 and 265/266: the authors mention that stabilizing substitutions for the RV1-F and NiV-F protein are based on the RV3 protein. It would benefit the reader if the mutations shown in Figure 6D and 6I could be matched to the RV3-F substitutions. From the manuscript, this is only possible for S41P.

We agree with the reviewer that this would make interpretation of the results easier. Since we do not want to overcrowd the figures, we choose instead to add explanations regarding the equivalent positions in the legends of Fig 6 for both RV1 and NiV.

7. Lines 267/268: Panel J is mentioned before panel I.

Good catch, we have swapped the statement in the text.

8. Lines 294/295: 'higher levels' of antibodies compared to mock suggests that mock animals had antibodies, which is not the case. This should be re-phrased.

Agreed, we have rephrased the statement as follows: " All mice generated preF and postF binding antibody titers above background of the mock control." This revised phrasing aims to highlight the noticeable increase in binding antibody titers across all the mice as compared to the control group.

9. Line 321: EctoOnly should be OnlyEcto.

Thank you for bringing this mistake to our attention; we have made the necessary correction.

10. Lines 324-327 and methods section: The pre-exposure of cotton rats with RV3 is not described.

In both the results and methods section, we made an addition to the statement to provide clarity on the experimental design and added: "Additionally, a control group was intranasally exposed to live RV3 at day 0." In the main text and "or were intranasally exposed to live RV3 (10^4 pfu/animal, VR-93TM; ATCC) at day 0" in the material and Method section.

11. Lines 413 and 414: 'protection' should be complemented with 'from infection'. The authors did not show pathology data for the upper or lower respiratory tract.

We have implemented the suggested improvement accordingly.

12. Lines 587-598: How were pre-challenge blood samples from cotton rats obtained?

The blood samples were collected via the retro-orbital route. We have added this information to the materials and method section (line 622 of the revised manuscript).

13. In Figure 1C, the authors should define a cut-off to define their most favorable combinations of mutations. This would help the reader to follow why for example S41P + F335 was not initially selected despite a beneficial effect of the combination.

While we believed F335P was not directly necessary for testing the feasibility of HRB stem optimization in Fig 2, we did add this substitution in the EctoOnly2P candidate. See also response and changes in response to major comment 1D above.

14. Figure 4 would benefit from figure legends. In which panel does the reader look at OnlyEcto2P or OnlyEcto data?

Excellent suggestion; we now have added legends within the figures for all SEC(-MALS) and DSF data of Fig 4.

15. Figure 4J: Should the mutations not be reversed? Δ S41P suggests that the difference from S to P is presented here but the panel shows the difference after reversion from P to S.

We plotted here the data of OnlyEcto and single revertants. For example, with Δ S41P we mean that the stabilizing S41P substitution is absent (thus position 41 has the wildtype Ser residue) from that specific construct. We were concerned that writing the actual substitution, e.g. P41S, would be confusing to readers, but are happy to change this based on the recommendation of the reviewer.

16. Figure 5A is not referred to in the manuscript.

Indeed, we only referred to fig 5A in the discussion. We now have included an additional reference at the beginning of the structure paragraph (line 212).

17. Figure 6 would benefit from figure legends. In which panel does the reader look at RV1 and where at Nipah virus? For the SEC data, the corresponding F substitutions could be mentioned.

The corresponding F substitutions between RV1/NiV and RV3 for analytical SEC are now mentioned in the figure legends instead, since we are afraid of overcrowding the figures with too much repetitive text. In the final, formatted paper the legends will be next to the figure and we believe it should be sufficiently clear for the readers which figures detail RV1 F and which NiV F.

18. sFig7A-C data could be added to Figure 6 of the main manuscript.

Indeed, during preparation of the manuscript we also contemplated whether to add these data to the main figures. In the end we did not because K338P's RV3 equivalent (F335P) was not used in the final RV3 preF lead candidate, and also in RV1 F it would be surface exposed so, though translatable, it is not among our favorite substitutions to use in a potential RV1 vaccine.

19. Figure 7A,B, E, G: the abbreviation Rel Pot titer should be explained in the caption.

We acknowledge that the previous axis label "Rel Pot" (relative potency) was not clear. We have changed it to "Binding IgG titer" to accurately describe the data.

20. Figure 7C, F, H and methods section: How was the IC50 titer determined? Was this a dose-response curve and would VN titer not be the more appropriate axis label?

In response to your question, we have incorporated a clarifying sentence into the materials and methods section, specifically addressing the calculation process of the IC50 titer: "VNA titers were

calculated as the serum dilution that caused a 50% reduction in GFP signal and expressed as log₂ IC₅₀ titer." We hope that this addition provides a clearer understanding of the methodology utilized. Additionally, we appreciate the suggestion to enhance the clarity of the axis labels. Consequently, we have made the necessary adjustment by replacing "IC₅₀ titers" with "Neutralizing antibody titer" on the axis labels.

21. Figure 7I and J would benefit from the figure legend from panel K.

In response to your feedback, we have made the necessary update and included the figure legend for both Figure 7I and 7J.

22. In both the txt and figures, sometimes the authors refer to viruses (for example RV3), sometimes to diseases (measles and nipah, instead of MeV and NiV). The authors should be consistent.

Thank you for pointing this out, we have made the necessary adjustments throughout the manuscript and now consistently use RV1 F and NiV F where appropriate.

REVIEWERS' COMMENTS

Reviewer #1 (Remarks to the Author):

All the remarks have been addressed and the clarity of the manuscript has improved. I have no further comments.

Reviewer #2 (Remarks to the Author):

The authors addressed the comments of the reviewers to satisfaction Concerning point 1 of reviewer 2, I would prefer this clarification to be taken up in the main text.

Reviewer #3 (Remarks to the Author):

In this revised version of their manuscript, Langedijk et al addressed many of my comments in the response-to-reviewers letter. Although they did not adopt most of my suggestions, like moving the in vivo data to before the RV1 and NiV structural data or adding more fusion data, I am happy with their reasoning for not doing this.

I have one minor comment left:

1. The cryoEM was performed with OnlyEcto2P and not OnlyEcto. The authors addressed this in the rebuttal, but the readers of the manuscript remain uninformed. A clarifying addition in the paragraph 'Structure of Stabilized RV3 preF' should be considered.